# Tuning site-specific dynamics to drive allosteric activation in a pneumococcal zinc uptake regulator

Daiana A Capdevila[1], Fidel Huerta[1,2], Katherine A Edmonds[1], My Tra Le[1], Hongwei Wu[1], David P Giedroc[1,3]*

[1]Department of Chemistry, Indiana University, Bloomington, United States;
[2]Graduate Program in Biochemistry, Indiana University, Bloomington, United States;
[3]Department of Molecular and Cellular Biochemistry, Indiana University, Bloomington, United States

**Abstract** MarR (multiple antibiotic resistance repressor) family proteins are bacterial repressors that regulate transcription in response to a wide range of chemical signals. Although specific features of MarR family function have been described, the role of atomic motions in MarRs remains unexplored thus limiting insights into the evolution of allostery in this ubiquitous family of repressors. Here, we provide the first experimental evidence that internal dynamics play a crucial functional role in MarR proteins. *Streptococcus pneumoniae* AdcR (adhesin-competence repressor) regulates $Zn^{II}$ homeostasis and $Zn^{II}$ functions as an allosteric activator of DNA binding. $Zn^{II}$ coordination triggers a transition from somewhat independent domains to a more compact structure. We identify residues that impact allosteric activation on the basis of $Zn^{II}$-induced perturbations of atomic motions over a wide range of timescales. These findings appear to reconcile the distinct allosteric mechanisms proposed for other MarRs and highlight the importance of conformational dynamics in biological regulation.
DOI: https://doi.org/10.7554/eLife.37268.001

*For correspondence:
giedroc@indiana.edu

## Introduction

Successful bacterial pathogens respond to diverse environmental insults or changes in intracellular metabolism by modulating gene expression (*Alekshun and Levy, 2007*). Such changes in gene expression are often mediated by 'one-component' transcriptional regulators, which directly sense chemical signals and convert such signals into changes in transcription. Members of the multiple antibiotic resistance regulator (MarR) family are critical for the survival of pathogenic bacteria in hostile environments, particularly for highly antibiotic-resistant pathogens (*Ellison and Miller, 2006*; *Yoon et al., 2009*; *Weatherspoon-Griffin and Wing, 2016*; *Tamber and Cheung, 2009*; *Aranda et al., 2009*; *Grove, 2017*). Chemical signals sensed by MarRs include small molecule metabolites (*Deochand and Grove, 2017*), reactive oxygen species (ROS) (*Liu et al., 2017*; *Sun et al., 2012*) and possibly reactive sulfur species (RSS) (*Peng et al., 2017*). It has been proposed that evolution of new MarR proteins enables microorganisms to colonize new niches (*Deochand and Grove, 2017*), since species characterized by large genomes and a complex lifestyle encode many, and obligate parasitic species with reduced genome sizes encode few (*Pérez-Rueda et al., 2004*). Therefore, elucidating how new inducer specificities and responses have evolved in this ubiquitous family of proteins on what is essentially an unchanging molecule scaffold is of great interest, as is the molecular mechanism by which inducer binding or cysteine thiol modification allosterically regulates DNA operator binding in promoter regions of regulated genes.

Obtaining an understanding of how allostery has evolved in one-component regulatory systems (*Ulrich et al., 2005*; *Marijuán et al., 2010*), including MarR family repressors, requires a comprehensive analysis of the structural and dynamical changes that occur upon inducer and DNA binding (*Capdevila et al., 2017a*; *Tzeng and Kalodimos, 2013*; *West et al., 2012*; *Tzeng and Kalodimos, 2009*; *Capdevila et al., 2018*). For MarRs, several distinct allosteric mechanisms have been proposed, from a 'domino-like' response (*Bordelon et al., 2006*; *Gupta and Grove, 2014*; *Perera and Grove, 2010*) to ligand binding-mediated effects on asymmetry within the dimer (*Anandapadamanaban et al., 2016*), to oxidative crosslinking of *E. coli* MarR dimers into DNA binding-incompetent tetramers (*Hao et al., 2014*). While there are more than 130 crystal structures of MarR family repressors in different allosteric states (*Figure 1—figure supplement 1*), an understanding of the role of atomic motions and the conformational ensemble in MarRs is nearly totally lacking and what is known is based exclusively on simulations (*Anandapadamanaban et al., 2016*; *Sun et al., 2012*). Here, we provide the first experimental evidence in solution that internal dynamics play a crucial functional role in a MarR protein, thus define characteristics that may have impacted the evolution of new biological outputs in this functionally diverse family of regulators.

In the conventional regulatory paradigm, the binding of a small molecule ligand, or the oxidation of conserved ROS-sensing cysteines, induces a structural change in the homodimer that typically negatively impacts DNA binding affinity. This results in a weakening or dissociation of the protein-DNA complex and transcriptional derepression. Several reports provide evidence for a rigid body reorientation of the two α4 (or αR)-reading heads within the dimer (*Figure 1A–B*, *Figure 1—figure supplement 1*) (*Alekshun et al., 2001*; *Fuangthong and Helmann, 2002*; *Wilke et al., 2008*; *Chang et al., 2010*; *Liu et al., 2017*; *Deochand and Grove, 2017*; *Dolan et al., 2011*; *Deochand et al., 2016*). The generality of this simple paradigm is inconsistent with the findings that some MarR proteins share very similar static structures in the DNA binding competent and DNA binding-incompetent states (*Anandapadamanaban et al., 2016*; *Kim et al., 2016*; *Liguori et al., 2016*); furthermore, several DNA binding competent states have been shown to require a significant rearrangement to bind DNA (*Alekshun et al., 2001*; *Liu et al., 2017*; *Zhu et al., 2017b*; *Hao et al., 2014*; *Gao et al., 2017*; *Chin et al., 2006*; *Saridakis et al., 2008*). In fact, a comprehensive analysis of all available MarR family structures strongly suggests that the degree of structural reorganization required to bind DNA, characterized by a narrow distribution of α4-α4' orientations, is comparable whether transitioning from the DNA-binding incompetent *or* competent states of the repressor (*Figure 1C*, *Table 1*, *Figure 1—source data 1*). These observations strongly implicate a conformational ensemble model of allostery (*Motlagh et al., 2014*) (*Figure 1B–D*), where inducer sensing impacts DNA binding by restricting the conformational spread of the active repressor, as was proposed in a recent molecular dynamics study (*Anandapadamanaban et al., 2016*).

MarR proteins are obligate homodimers that share a winged-helical DNA-binding domain connected to a DNA-distal all-helical dimerization domain where organic molecules bind in a cleft between the two domains (*Figure 1—figure supplement 1B*). Individual MarR members have been shown to bind a diverse range of ligands at different sites on the dimer (*Otani et al., 2016*; *Takano et al., 2016*); likewise, oxidation-sensing cysteine residues are also widely distributed in the dimer (*Fuangthong and Helmann, 2002*; *Liu et al., 2017*; *Hao et al., 2014*; *Dolan et al., 2011*; *Chen et al., 2006*). This functional diversity is accompanied by relatively low overall sequence similarity, which suggests that a conserved molecular pathway that connects sensing sites and the DNA binding heads is highly improbable. Complicating our current mechanistic understanding of this family is that for many members, including *E. coli* MarR, the physiological inducer (if any) is unknown, rendering functional conclusions on allostery from crystallographic experiments alone less certain (*Hao et al., 2014*, *Zhu et al., 2017b*).

In contrast to the extraordinary diversity of thiol-based switching MarRs, MarR family metallosensors are confined to a single known regulator of $Zn^{II}$ uptake, exemplified by AdcR (adhesin competence regulator) from *S. pneumoniae* and closely related *Streptococcus ssp.* (*Loo et al., 2003*; *Reyes-Caballero et al., 2010*) and ZitR from *Lactococcus spp* (*Llull et al., 2011*; *Zhu et al., 2017c*). AdcR and ZitR both possess two closely spaced pseudotetrahedral $Zn^{II}$ binding sites termed site 1 and site 2 (*Figure 1A*) that bind $Zn^{II}$ with different affinities (*Reyes-Caballero et al., 2010*; *Guerra et al., 2011*; *Sanson et al., 2015*; *Zhu et al., 2017c*). $Zn^{II}$ is an allosteric *activator* of DNA operator binding which is primarily dependent on the structural integrity of site 1 (*Reyes-Caballero et al., 2010*; *Zhu et al., 2017c*). ZitR has been recently structurally characterized, with

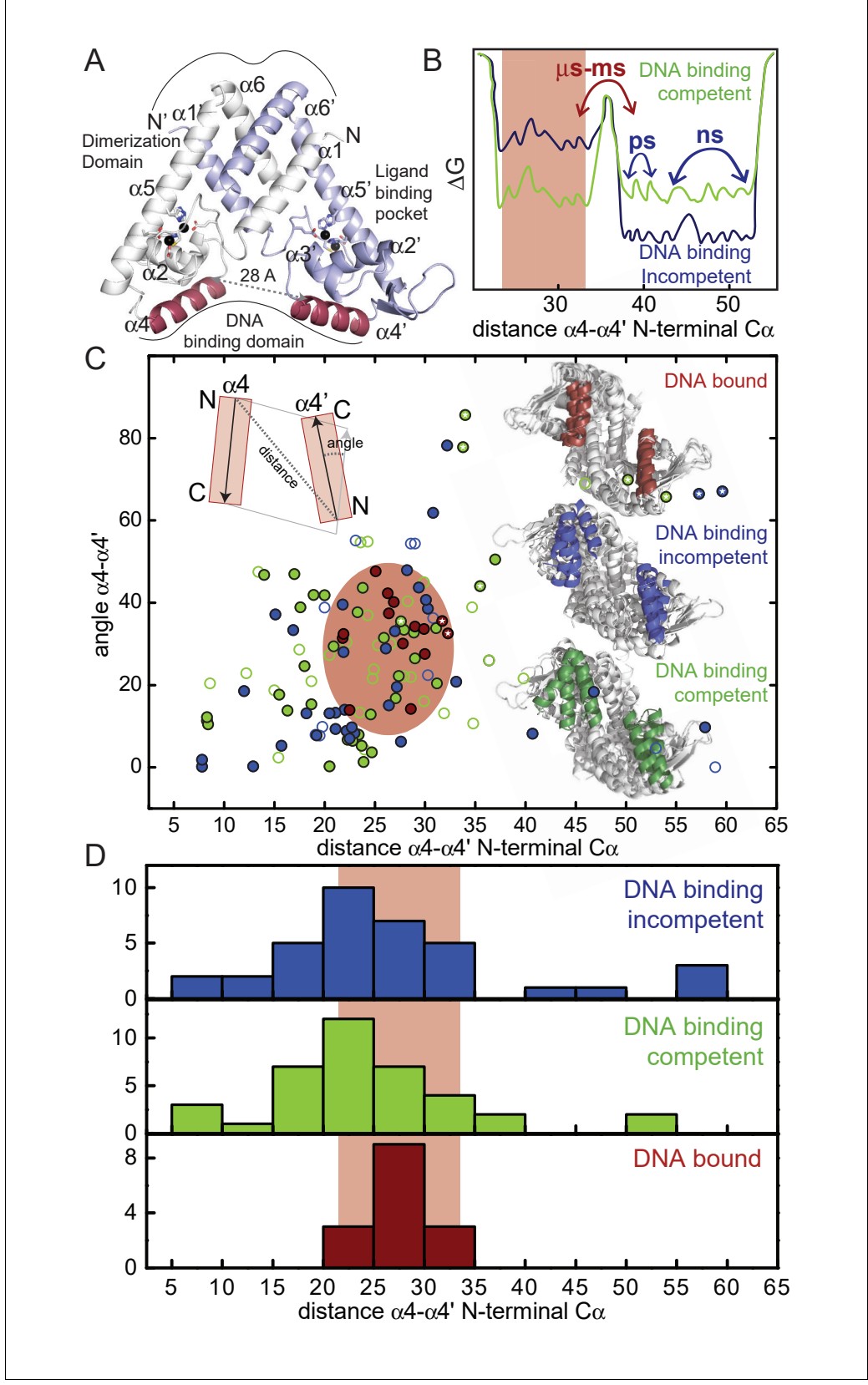

**Figure 1.** (A) Ribbon representation of dimeric Zn(II)-bound AdcR, with one protomer shaded white and the other shaded *light blue* (PDB code: 3tgn; *Guerra et al., 2011*). The two Zn(II) ions in each protomer are represented by

*Figure 1 continued on next page*

*Figure 1 continued*

spheres, and coordinating ligands are shown in stick representation. The DNA binding helices are shaded *red*. (B) Simplified free energy diagram showing the DNA binding competent (*green*) and DNA binding incompetent (*blue*) states with the relative population of two distinct conformations: compatible with DNA binding (*red* rectangle, α4-α4' distance between DNA binding helices, ≈ 30 Å) and incompatible with DNA binding (larger α4-α 4' distances). In this free energy diagram, the DNA binding-incompetent state has a comparatively higher population of the conformation incompatible with DNA binding relative to the DNA binding-competent state. (C) The α4-α4' distance distribution plotted against the DNA-binding inter-helical α4-α4' orientation distribution for all the reported MarR crystal structures (see *Table 1* and *Figure 1—source data 1* for details) in the allosterically DNA binding competent conformation (*green*), a DNA binding incompetent conformation (*blue*) and in the DNA-bound (*red*) conformation. *Filled* circles represent states that have been assigned based on DNA binding data, while for the *hollow* circles the DNA binding properties were assigned taking into account the conformational state in the crystal structure (*i.e.*, reduced, ligand bound) and the degree of sequence similarity to other MarR repressors. The structures for ZitR and AdcR have been highlighted with a *white* star. The inferred conformational space occupied by the DNA-bound conformation in all MarR regulators (*Table 1*) is shaded in *red* oval. Ribbon representations of the molecules in each conformation are shown in the inset, as well as a scheme of how the distances and angles were measured. (D) Histogram plot of the α4-α4' distance (see panel C) extracted from 136 different crystal structures of MarR repressors in the DNA binding incompetent, DNA binding competent and DNA-bound conformations.

DOI: https://doi.org/10.7554/eLife.37268.002

The following source data and figure supplement are available for figure 1:

**Source data 1.** Table of details on MarR proteins structures: PDB ID, reference for the structure, protein state and ligand bound (if any), DNA binding competence classification, organism, Methods used for determining the DNA binding properties, DNA binding constant ($K_a$), reference for the DNA binding constant, residues in the α1-α2 loop obtained from pymol secondary sequence assignment, minimal distance between the α4-α4' helices, and angle between the α4-α4' helices.

DOI: https://doi.org/10.7554/eLife.37268.004

**Figure supplement 1.** Structural comparison ZitR and AdcRs with other MarR family repressors.

DOI: https://doi.org/10.7554/eLife.37268.003

---

crystallographic models now available for the apo- and $Zn^{II}_1$- (bound to site 1) and $Zn^{II}_2$- and $Zn^{II}_2$-DNA operator complexes, thus providing significant new insights into ZitR and AdcR function (*Zhu et al., 2017c*). These structures reveal that $Zn^{II}_2$-ZitR and $Zn^{II}_2$-AdcR form triangularly-shaped homodimers and are essentially identical, as anticipated from their high sequence identity (49%). Apo-ZitR adopts a conformation that is incompatible with DNA binding, and filling of both $Zn^{II}$ sites is required to adopt a conformation that is similar to that of the DNA-complex. Thermodynamically, filling of the low affinity site two enhances allosteric activation of DNA-binding by ≈ 10-fold, and this occurs concomitant with a change in the H42 donor atom to the site 1 $Zn^{II}$ ion from Nε2 in the apo- and $Zn^{II}_1$-states to Nδ1 in the $Zn^{II}_2$-ZitR [as in $Zn^{II}_2$ AdcR; (*Guerra et al., 2011*) and $Zn^{II}_2$ ZitR-DNA operator complexes (*Zhu et al., 2017c*). Allosteric *activation* by $Zn^{II}$ is in strong contrast to all other members of the MarR superfamily, consistent with its biological function as uptake repressor at high intracellular $Zn^{II}$.

Here we employ a combination of NMR-based techniques and small angle x-ray scattering (SAXS) to show that apo- (metal-free) AdcR in solution is characterized by multiple semi-independent domains connected by flexible linkers, resulting in a distinct quaternary structure from the Zn-bound state previously structurally characterized (*Guerra et al., 2011*). Our backbone relaxation dispersion-based NMR experiments show that apo-AdcR samples distinct conformational states in the μs-ms timescale, while $Zn^{II}$ narrows this distribution, likely increasing the population of a state that has higher affinity for DNA. This finding is consistent with the crystallographic structures of $Zn^{II}_2$ ZitR and the $Zn^{II}_2$ ZitR:DNA complex (*Zhu et al., 2017c*). The site-specific backbone and methyl sidechain dynamics in the sub-ns timescale show that $Zn^{II}$ not only induces a general restriction of these internal protein dynamics, but also subtly enhances fast timescale backbone and sidechain motions in the DNA binding domains. Together, these data suggest that $Zn^{II}$ coordination drives a conformational change that enhances internal dynamics uniquely within the DNA binding domain, thus poising the repressor to interact productively with various DNA operator target sequences (*Reyes-Caballero et al., 2010*). We demonstrate the functional importance of these dynamics by

**Table 1.** Interprotomer distances between the Cα of the N-terminal residue in the α4 and α4' helices for representatives MarR proteins

| MarR | DNA-bound state | | DNA binding incompetent state[a],* | | DNA binding competent state[b]* | |
|---|---|---|---|---|---|---|
| | Distance (Å) | Pdb id | Distance (Å) | Pdb id | Distance (Å) | Pdb id |
| ZitR (AdcR) | 32.3/31.7 | 5yi2/5yi3 | 59.6/57.3 | 5yh0/5yh1 | 35.5/54.0/50.2 (22.2/34/33.8) | 5yhx/5yhy/ 5yhz (3tgn/5jls/5 jlu) |
| *Ec* MarR | 29 | 5hr3 | 12.9/12 | 1jgs/4jba | 8.3/8.4 | 3vod/3voe |
| OhrR | 27.6 | 1z9c | (32.2) | (2pfb) | 23.9 (28.9) | 1z91 (2pex) |
| SlyA | 27.8 | 3q5f | 29.4 | 3deu | 15.5 (23.8, 20) | 3qpt (1lj9, 4mnu) |
| AbsC | 26.3 | 3zpl | 30.8 | 3zmd | - | - |
| RovA | 21.8/21.9 | 4aij/ 4aik | - | - | 20.9 | 4aih |
| MosR | 25.1 | 4f×4 | 15.1 | 4f×0 | - | - |
| MepR | 26.4/26.9 | 4lll/ 4lln | 18.9/16.9/ 30.8/57.9 | 3eco/4l9n/ 4l9t/4l9v | 27.9/46.8 | 4l9j/4ld5 |
| AbfR | 29.9/30 | 5hlh/5hlg | 40.7 | 5hli | 37 | 4hbl |
| Rv2887 | 22.5 | 5hso | 7.9/15.1 | 5hsn/5hsl | 8.3 | 5hsm |
| HcaR | 28.6 | 5bmz | 19.1/19.8/19.5/19.2 | 4rgx/4 rgu/4rgs/ 4rgr | 18.7 | 3k0l |
| ST1710 | 10.1[c] | 3gji | 23 | 3gf2 | 22.8 | 2eb7 |
| TcaR | 19.1[d] | 4kdp | 22.3/24.7 | 4eju/3kp7 | 26.4/22.5/21.1/22/27.6/ 18.3/21.1/18.2 | 3kp2/3kp3/3kp4/3kp5 /3kp7/4ejt/4ejv/4ejw |

[a]Any protein allosteric state that has been shown to bind to DNA *in-vitro* with an affinity higher than $10^7$ M$^{-1}$ or is capable of repressing the expression of downstream gene.

[b]Any protein allosteric state that fails to repress these genes and/or exhibits a significantly lower DNA binding affinity from the DNA binding-competent conformation (at least 10-fold) or an affinity lower than $10^6$ M$^{-1}$ *In addition to these two categories, two other categories were classified as DNA binding-competent or DNA binding-incompetent states in **Figure 1C**. They refer to any protein allosteric state for which the DNA binding properties have not been determined, but the conformational state in the crystal structure is known (i.e., reduced, ligand bound).

[c]Not inserted in the major groove of the DNA.

[d]This structure was co-crystallized with ssDNA. Any entry in parentheses corresponds to a structure of a homologue from a different organism (see **Figure 1—source data 1**).

DOI: https://doi.org/10.7554/eLife.37268.005

characterizing both methyl sidechain and hydrogen-bonding substitution mutants of AdcR (**Capdevila et al., 2017a**) in terms of function, stability and dynamical impact. Overall, our findings suggest that protein dynamics on a wide range of timescales strongly impact AdcR function. We propose an ensemble model of allostery that successfully reconciles the distinct mechanisms proposed for other MarR family repressors and suggests a mechanism of how evolution tunes dynamics and structure to render distinct biological outputs (allosteric activation vs. allosteric inhibition) on a rigorously conserved molecular scaffold.

## Results and discussion

### Solution structural differences between apo and Zn[II] bound forms of AdcR

Our crystal structure suggests that once AdcR is bound to both Zn[II], the αR- (α4) reading heads adopt a favorable orientation for DNA binding (**Guerra et al., 2011**), a finding compatible with structural studies of *L. lactis* ZitR (**Zhu et al., 2017c**) (**Figure 1A**). These structural studies suggest a 'pre-locked' model, where Zn[II] binding to both sites 1 and 2, concomitant with a H42 ligand atom switch, locks the AdcR homodimer into a DNA binding-competent conformation. This model makes the prediction that the unligated AdcR can explore conformations structurally incompatible with

DNA binding, as shown previously for $Zn^{II}_1$ ZitR (*Zhu et al., 2017c*), thus requiring a significant degree of reorganization to bind with high affinity to the DNA (*Figure 1B*). Despite substantial efforts, it has not yet been possible to obtain the crystal structure of apo-AdcR, suggesting that the apo-repressor may be highly flexible in solution (*Guerra et al., 2011*; *Sanson et al., 2015*). Thus, we employed SAXS as a means to explore the apo-AdcR structure and elucidate the structural changes induced by $Zn^{II}$ binding and conformational switching within the AdcR homodimer.

We first examined the behavior of apo- and $Zn^{II}$-bound states. Both states show Guinier plots indicative of monodispersity and similar radii of gyration ($R_g$). These data reveal that each state is readily distinguished from the other in the raw scattering profiles (to $q = 0.5$ Å$^{-1}$) (*Figure 2A*)as well as in the PDDF plots (*p(r) versus r*), with the experimental scattering curve of the $Zn^{II}$ bound state being more consistent than the unligated state with the one obtained from the $Zn^{II}_2$ AdcR crystal

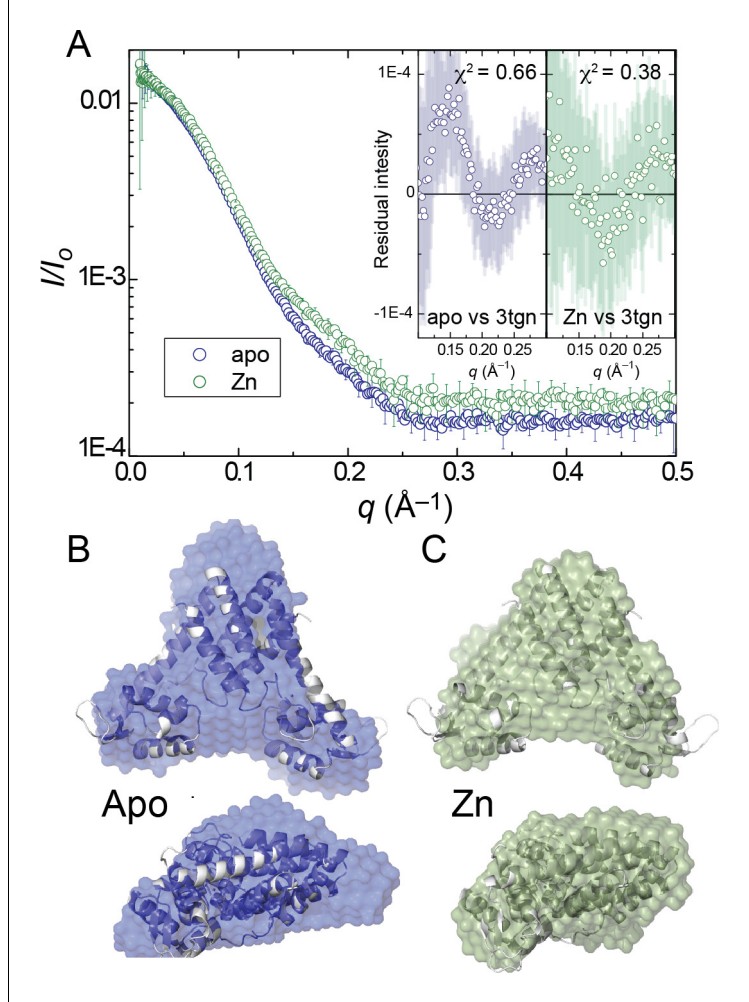

**Figure 2.** (A) Small angle X-ray scattering (SAXS) curve of AdcR in apo- and $Zn_2$-states. Insets present the residual intensity and $\chi^2$ estimated for the calculated scattering profile of the previously published AdcR-$Zn_2$ structure (PDB: 3tgn) in comparison with the scattering profiles of AdcR of apo and $Zn_2$-states (*Guerra et al., 2011*). Best-fit DAMMIF *ab initio* model (*Franke and Svergun, 2009*) for apo- (B) (*blue*) and $Zn^{II}_2$-states (C) (*green*), aligned with the ribbon representation of the $Zn^{II}_2$ structure (*Figure 1A*, PDB: 3tgn). The corresponding Guinier, Kratky and pairwise distribution histogram plots are shown in *Figure 2—figure supplement 1*, along with the fitting parameters.

DOI: https://doi.org/10.7554/eLife.37268.006

The following figure supplement is available for figure 2:

**Figure supplement 1.** Small angle X-ray scattering (SAXS) analysis of AdcR in the apo and Zn-binding states.
DOI: https://doi.org/10.7554/eLife.37268.007

structure (*Figure 2A*, inset). Moreover, a qualitative analysis of the PDDF plots suggests that apo-AdcR is less compact than the $Zn^{II}$-bound state (*Figure 2—figure supplement 1*). The molecular scattering envelopes calculated as bead models with the *ab initio* program DAMMIF for apo-AdcR suggest that the differences between the apo and $Zn^{II}$ AdcR SAXS profiles can be explained on the basis of a reorientation of the winged helix-turn-helix motif with respect to the dimerization domain, particularly in a distortion in the α5 helix (*Figure 2B*). The models obtained confirm that the Zn-bound structure in solution resembles the crystallographic models of apo-ZitR and $Zn^{II}$ AdcR (*Guerra et al., 2011*; *Zhu et al., 2017c*) (*Figure 2C*); however, we note that the SAXS profile of the apo-AdcR differs significantly from the ZitR crystal structure (*Figure 2—figure supplement 1D*) which is likely related to the high flexibility of this conformational state in solution. Moreover, the resolution of SAXS based models cannot be used to obtain residue-specific information about structural perturbations introduced by $Zn^{II}$ binding (*Figure 2—figure supplement 1*). Thus, we turned to NMR-based techniques to provide both high resolution and site-specific information on this highly dynamic system.

TROSY NMR on the 100% deuterated AdcR homodimer (32 kDa) and optimized buffer conditions for both states (pH 5.5, 50 mM NaCl, 35°C) enabled us to obtain complete backbone assignments for $Zn^{II}_2$-AdcR and nearly complete assignments for apo-AdcR (missing residues 21, 38 – 40 due to exchange broadening) (*Figure 3—figure supplement 1*). The chemical shift perturbation maps (*Figure 3A–B*) reveal that the largest perturbations are found in the immediate vicinity of the metal site region, that is the α1-α2 loop (residues 21 – 35), the remainder of the α2 helix (residues 41 – 47), and the central region of the α5 helix, which provides donor groups to both site 1 (H108, H112) and site 2 (E107) $Zn^{II}$. These changes derive partially from changes in secondary structure, such as the extension of the α1 helix and partial unfolding of the α2 helix (*Figure 3—figure supplement 1*), as well as from proximity to the $Zn^{II}$.

The changes in Cα and Cβ chemical shifts in the central region of the α5 helix and the presence of strong NOEs to water for these residues are consistent with a kink in this helix in the apo-state (*Figure 3—figure supplement 2A–B*), as is commonly found in other structurally characterized MarR repressors in DNA-binding incompetent conformations (*Zhu et al., 2017b*; *Duval et al., 2013*). However, the kink is expected to be local and transient, since a TALOS+ analysis of chemical shifts predicts that the α5 helix remains the most probable secondary structure for all tripeptides containing these residues in the apo-state (*Shen et al., 2009*) (*Figure 3—figure supplement 2C*). The backbone changes in chemical shifts are accompanied by changes in the hydrophobic cores in the proximity of $Zn^{II}$ binding as reported by the stereospecific sidechain methyl group chemical shift perturbation maps (*Figure 3B*). Comparatively smaller perturbations extend to the α1 helix and the C-terminal region of the α6 helix, DNA-binding α4 helix (S74) and into the β-wing itself, consistent with a significant change in quaternary structure within the AdcR homodimer upon binding of both allosteric metal ions (*Figure 3A–B*).

Overall, our NMR and SAXS data show that the main structural differences between the apo- and $Zn^{II}_2$ states are localized in the region immediately surrounding the $Zn^{II}$ coordination sites, giving rise to a change in quaternary structure, while conserving the size and the overall secondary structure of the molecule. In particular, our data point to a kink in the α5 helix and a structural perturbation in the α1-α2 loop, which could be inducing a reorientation of the winged helix-turn-helix motifs relative to the dimerization domain. In addition to these structural changes, metal binding seems to be restricting the α1-α2 loop dynamics by means of metal coordination bonds, a hydrogen-bond network (*Chakravorty et al., 2013*) and other intermolecular contacts within the dimerization and DNA binding domains (*Zhu et al., 2017c*). Flexibility of the α1-α2 loop could potentially destabilize the DNA complex; in this case, interactions formed as a result of $Zn^{II}$ coordination may be important in allosteric activation of DNA binding. Such a dynamical model contrasts sharply with a rigid body mechanism as previously suggested for other MarRs (*Alekshun et al., 2001*; *Chang et al., 2010*; *Dolan et al., 2011*; *Saridakis et al., 2008*; *Birukou et al., 2014*; *Radhakrishnan et al., 2014*), thus motivating efforts to understand how conformational dynamics impacts biological regulation by $Zn^{II}$ in AdcR.

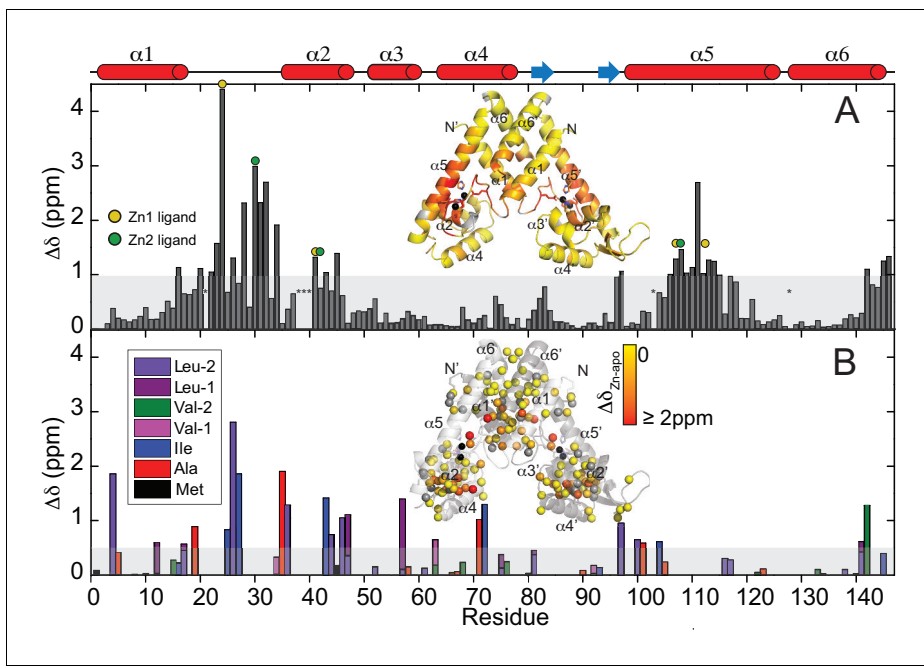

**Figure 3.** Chemical shift perturbation (CSP) maps for Zn[II] binding to to AdcR. (**A**) Backbone CSPs. CSPs of the sterospecifically assigned methyl groups at pH 5.5, 50 mM NaCl, 35°C. (**B**) Both these CSPs are painted on the ribbon representation of the structure of Zn[II]$_2$ AdcR. The shaded bar in each case represents one standard deviation from the mean perturbation. Site 1 and site 2 ligands in the primary structure in panel A are denoted by the *yellow* and *green* circles, respectively; the asterisks at residue positions 21 and 38 – 40 indicate no assignment in the apo-state (see materials and methods), while asterisks mark residue positions 103 and 128 for prolines. Insets show the CSP values painted onto the 3tgn structure.

DOI: https://doi.org/10.7554/eLife.37268.008

The following figure supplements are available for figure 3:

**Figure supplement 1.** Backbone [1]H-[15]N TROSY spectra showing the assignments of (**A**) apo WT AdcR and (**B**) Zn[II] WT AdcR.

DOI: https://doi.org/10.7554/eLife.37268.009

**Figure supplement 2.** NMR analysis of the α5 helix in AdcR.

DOI: https://doi.org/10.7554/eLife.37268.010

## Zn[II]-induced changes in AdcR conformational plasticity along the backbone

We therefore turned to an investigation of protein dynamics in AdcR. [15]N $R_1$, $R_2$, and steady-state heteronuclear [15]N{[1]H} NOEs provide information on internal mobility along the backbone, as well as on the overall rotational dynamics (*Figure 4A–D*; *Figure 4—figure supplements 1–2*). The $R_1$ and $R_2$ data reveal that Zn[II]$_2$ AdcR tumbles predominantly as a single globular unit in solution with a rotational diffusion tensor and [15]N $R_2/R_1$ ratio compatible with those parameters predicted from the crystal structure (*Guerra et al., 2011*) using hydroNMR (*García de la Torre et al., 2000*) (*Figure 4B*; *Figure 4—figure supplement 1*). The β-wing region tumbles independently from the rest of the molecule (*Figure 4B*, *Figure 4—figure supplement 1B*). These data also reveal that the α1-α2 linker region that donates the E24 ligand to Zn[II] binding site one is ordered to an extent similar to the rest of the molecule (*Figure 4—figure supplement 1B*). In striking contrast, in apo-AdcR, the dimerization and DNA-binding domains each have a significantly smaller [15]N $R_2/R_1$ ratio (*Figure 4B*), somewhat closer to what is expected if these domains tumble independently of one another in solution, which might be facilitated by a highly dynamic α1-α2 loop (see also *Figure 4—figure supplement 1*). These findings are consistent with the SAXS data, which show that apo-AdcR is less compact than the Zn[II]$_2$ state. As in the Zn[II]$_2$ state, the β−wing tumbles independently of the rest of the molecule, revealing that a change in the flexibility or orientation of the β−hairpin is likely not part of the allosteric mechanism, contrary to what has been proposed for other MarRs on the basis of crystal

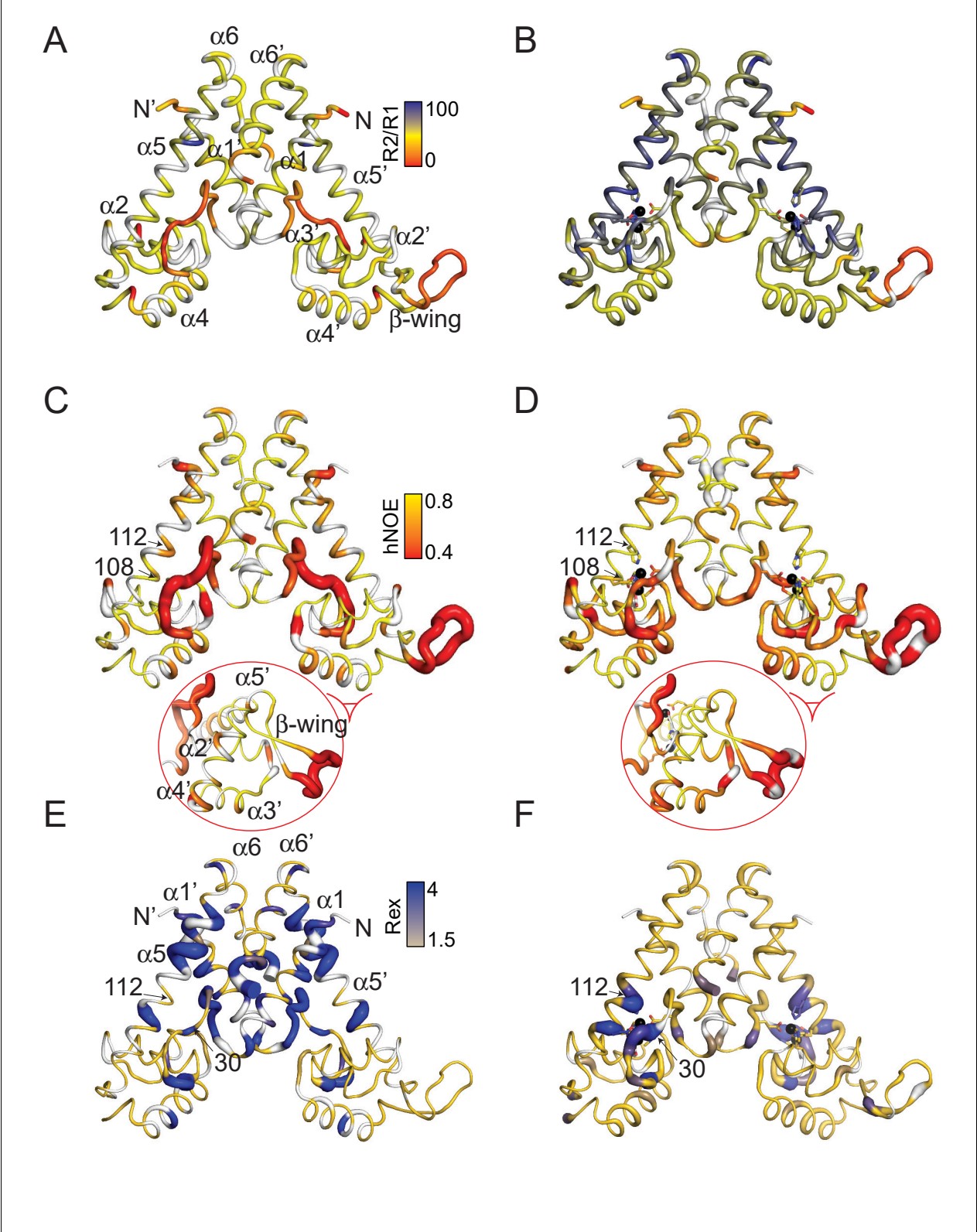

**Figure 4.** Dynamical characterization of the apo- (A) (C) (E) and Zn$^{II}_2$ (B) (D) (F) AdcR conformational states. Backbone $^1$H-$^{15}$N amide $R_2/R_1$ for apo- (A) and Zn$^{II}_2$ AdcR (B) painted onto the 3tgn structure (*Guerra et al., 2011*). Heteronuclear NOE analysis of apo- (C) and Zn$^{II}_2$ (D) AdcR with the values of the $^{15}$N-{$^1$H}-NOE (hNOE) painted onto the 3tgn structure. Values of $R_{ex}$ determined from HSQC $^{15}$N-$^1$H CPMG relaxation dispersion experiments at a field of 600 MHz for the apo- (E) and Zn$^{II}_2$ (F) AdcRs (see *Figure 4—figure supplement 3* for complete data sets). Similar results were obtained at 800

*Figure 4 continued on next page*

*Figure 4 continued*

MHz. Zn$^{II}$ ions are shown as black spheres and residues excluded due to overlap are shown in gray. The width of the ribbon reflects the value represented in the color bar.

DOI: https://doi.org/10.7554/eLife.37268.011

The following figure supplements are available for figure 4:

**Figure supplement 1.** Backbone $^{15}$N spin relaxation analysis of AdcR.

DOI: https://doi.org/10.7554/eLife.37268.012

**Figure supplement 2.** Dynamic parameters obtained for (**A**) apo- and (**B**) Zn$^{II}$$_2$-WT AdcR using tensor 2.

DOI: https://doi.org/10.7554/eLife.37268.013

**Figure supplement 3.** Backbone (NH) relaxation dispersion analysis of AdcR.

DOI: https://doi.org/10.7554/eLife.37268.014

structures alone (*Liu et al., 2017*; *Deochand and Grove, 2017*; *Kim et al., 2016*). Overall, the $^{15}$N relaxation data for backbone amides suggest that Zn$^{II}$ binding leads to a reduction of mobility of the α1-α2 loop, which in turn, decreases the dynamical independence the DNA-binding and dimerization domains, thereby stabilizing a conformation that tumbles in solution as a single globular unit.

To further probe this reduction of flexibility upon Zn$^{II}$ binding, we investigated sub-nanosecond backbone mobility as reported by the steady-state heteronuclear $^{15}$N{$^1$H} NOEs (*Figure 4C–D*, *Figure 4—figure supplement 1*, *Figure 4—figure supplement 2*) and millisecond mobility as reported by $^{15}$N relaxation dispersion experiments (*Figure 4E–F*, *Figure 4—figure supplement 3*). These hNOE data confirm that the internal mobility of the apo-state on this timescale largely localizes to the β−wing, the α1-α2 loop, and the central region of the α5 helix, around E107 (Zn$^{II}$ site 2 ligand) and H108 and H112 (Zn$^{II}$ site 1 ligands) (*Figure 4C*, *Figure 4—figure supplement 1*). This short-timescale flexibility in these regions is significantly restricted upon Zn$^{II}$ binding, but somewhat paradoxically leads to a small *increase* in sub-nanosecond backbone motion in the DNA-binding domain (*Figure 4C–D*, inset), particularly in the α2 helix, the α3 helix and the N-terminal region of the α4 helix, the latter of which harbors the key DNA-binding determinants (*Figure 1—figure supplement 1A*) (*Zhu et al., 2017c*). The Zn$^{II}$- induced quenching of sub-nanosecond mobility is also accompanied by an increase in mobility on the µs-ms (slow) timescale in the metal binding site, particularly at or near metal binding residues, including H112 (site 1) and C30 (site 2) (*Figure 4F*). In addition, the slow timescale backbone dynamics show a restriction of a conformational sampling in a band across the middle of the dimerization domain, including the upper region of the α5 helix, the N-terminus of α1, and the C-terminus of α6 (*Figure 4E–F*). These slow motions in the apo-state likely report on a global breathing mode of the homodimer reflective of the conformational ensemble, which is substantially restricted upon Zn$^{II}$ binding.

These large differences in structure and dynamics between the apo and Zn$^{II}$$_2$ AdcRs along the backbone suggest an allosteric mechanism that relies on a redistribution of internal mobility in both fast- and slow timescales, rather than one described by a rigid body motion. This mobility redistribution effectively locks AdcR in a triangular shape compatible with DNA binding, while also inducing a small, but measurable increase in motional disorder in the DNA binding domain (*Figure 4C–D*). Since other studies connect changes in motional disorder like these to sequence recognition and high affinity binding to DNA, particularly in the side chains (*Capdevila et al., 2017a*; *Kalodimos et al., 2004*; *Anderson et al., 2013*), we decided to probe side chain dynamics in greater detail.

## Zn$^{II}$-induced perturbations of side chain conformational disorder in AdcR

Sub-nanosecond timescale dynamics have been used as a proxy for the underlying thermodynamics of ligand binding and can report on the role of conformational entropy ($\Delta S_{conf}$) in allosteric mechanisms (*Caro et al., 2017*; *Frederick et al., 2007*; *Sharp et al., 2015*) The contribution of changes in backbone dynamics to the $\Delta S_{conf}$ of ligand binding processes measured in a number of model systems has been shown to be small (<5%), relative to the contribution to $\Delta S_{conf}$ from the side chains (*Caro et al., 2017*). However, in the case of AdcR, Zn$^{II}$ binding clearly restricts the backbone dynamics of the α1-α2 loop as reflected by an increase in the N-H order parameters in this region ($S^2_{bb}$, *Figure 4—figure supplement 2*), which sums to $-T\Delta S_{conf,\ bb}$ to ≈3.5 kcal mol$^{-1}$ (see materials and

methods). Thus, α1-α2 loop restriction to the internal dynamics may well be a significant contributor to the underlying thermodynamics of metal binding. Moreover, if this motional redistribution along the backbone is accompanied by changes in the internal dynamics of the side chains, particularly those in the DNA binding domain, these fast internal dynamics could greatly impact the entropy of metal binding and/or allostery. Mapping these perturbations by measuring the change in methyl group order parameter ($\Delta S^2_{axis}$) upon $Zn^{II}$ binding, employed as dynamical proxy (*Capdevila et al., 2017a*; *Caro et al., 2017*) may in turn, pinpoint residues with functional roles, that is allosteric hot-spots (*Capdevila et al., 2017a*; *Capdevila et al., 2018*).

We measured the axial order parameter, $S^2_{axis}$, for all 82 methyl groups, comparing the apo- and Zn-bound states of AdcR (*Figure 5—figure supplement 1*). These dynamics changes are overall consistent with the stiffening observed along the protein backbone, for example in the α1-α2 loop; L26, in particular, is strongly impacted, changing motional regimes, $|\Delta S^2_{axis}|>0.2$ (*Frederick et al., 2007*) (*Figure 5A*). We observe a significant redistribution of sidechain mobility throughout the molecular scaffold (23 probes change motional regimes), as has been previously shown for other transcriptional regulators (*Capdevila et al., 2017a*; *Tzeng and Kalodimos, 2012*), summing to a small net decrease in conformational entropy upon $Zn^{II}$ coordination, $-T\Delta S_{conf,sc} = 1.1 \pm 0.2$ kcal mol$^{-1}$ (*Figure 5B*). Note that this value is quantitatively less than that attributed to the backbone of the α1-α2 loop. However, many of the methyl groups that change motional regimes are located in the DNA binding domain (*Figure 5A–B*, *Figure 5—figure supplement 2*). In particular, the side chain flexibility of many residues in the α3 helix *increases*, including L47, L57, L61, while a small hydrophobic core in the C-terminus of the α4 helix stiffens significantly, for example L81, V34. These changes are accompanied by perturbations in the dynamics at the dimer interface, that is L4, I16, V142, in both motional regimes as reported by $\Delta S^2_{axis}$ and $\Delta R_{ex}$ (in the µs-ms timescale), the latter derived from relaxation dispersion experiments (*Supplementary file 1*-Table S1; *Figure 5—figure supplement 3*).

## On-pathway and off-pathway allosterically impaired mutants of AdcR

Our previous work (*Capdevila et al., 2017a*) makes the prediction that 'dynamically active' side-chains (methyl groups with $|\Delta S^2_{axis}|>0.1$ upon $Zn^{II}$ binding) (see *Figure 5A–B*) are crucial for allosteric activation of DNA binding by $Zn^{II}$. To test this prediction, we prepared and characterized several mutant AdcRs in an effort to disrupt allosteric activation of DNA binding, while maintaining the structure and stability of the dimer, and high affinity $Zn^{II}$ binding. Since it was not clear *a priori* how mutations that perturb mobility distributions in one timescale or the other (sub-ns or µs-ms) would impact function, we focused on two kinds of substitution mutants: methyl group substitution mutants of dynamically 'active' side chains positioned in either the DNA binding or the dimerization subdomains (*Figure 6A,B*) (*Capdevila et al., 2017a*), and substitutions in the hydrogen-bonding pathway in the Zn-state that may contribute to the rigidity of the α1-α2 loop in $Zn^{II}_2$-AdcR (*Figure 6A*) (*Chakravorty et al., 2013*). We measured DNA binding affinities of the apo and $Zn^{II}_2$-states, and calculated the allosteric coupling free energy, $\Delta G_c$, from $\Delta G_c=-RT\ln(K_{Zn,DNA}/K_{apo,\ DNA})$ (*Giedroc and Arunkumar, 2007*) (*Figure 6C*, *Figure 6—figure supplement 1* and *Table 2*). All mutants are homo-dimers by size-exclusion chromatography (*Figure 6—figure supplement 2*) and all bind the first pro-tomer equivalent of $Zn^{II}$ (to site 1) with wild-type-like affinity (*Figure 6—figure supplement 3*, *Supplementary file 1*-Table S1). Two of the sixteen mutants investigated here (L61V and V63A AdcRs) showed a significantly lower thermal stability as estimated by differential scanning fluorimetry (*Figure 6—figure supplement 4*, *Supplementary file 1*-Table S2); this prevented a quantitative analysis of their DNA and metal binding affinities and thus they were not considered further.

## DNA-binding domain mutants

The redistribution of fast time scale side-chain dynamics in the DNA binding domain is delocalized throughout the different secondary structure motifs (*Figure 5A–B*). Thus, we prepared several methyl substitution mutants of methyl-bearing residues in the α3 (L57, L61, V63), α4 (L81) and α5 (I104) helices, as well as two residues in the α1-α2 loop, V34 and L36. I104 and V63 are not dynamically active in AdcR ($|\Delta S^2_{axis}|<0.1$; $\Delta R_{ex} <1.0$); thus, these mutant are predicted to function as control substitutions. V34 and L36 are dynamically active on both timescales, which is not surprising since the α1-α2 loop folds upon $Zn^{II}$ binding to AdcR (*vide supra*) (*Zhu et al., 2017c*). In contrast, L57, L61

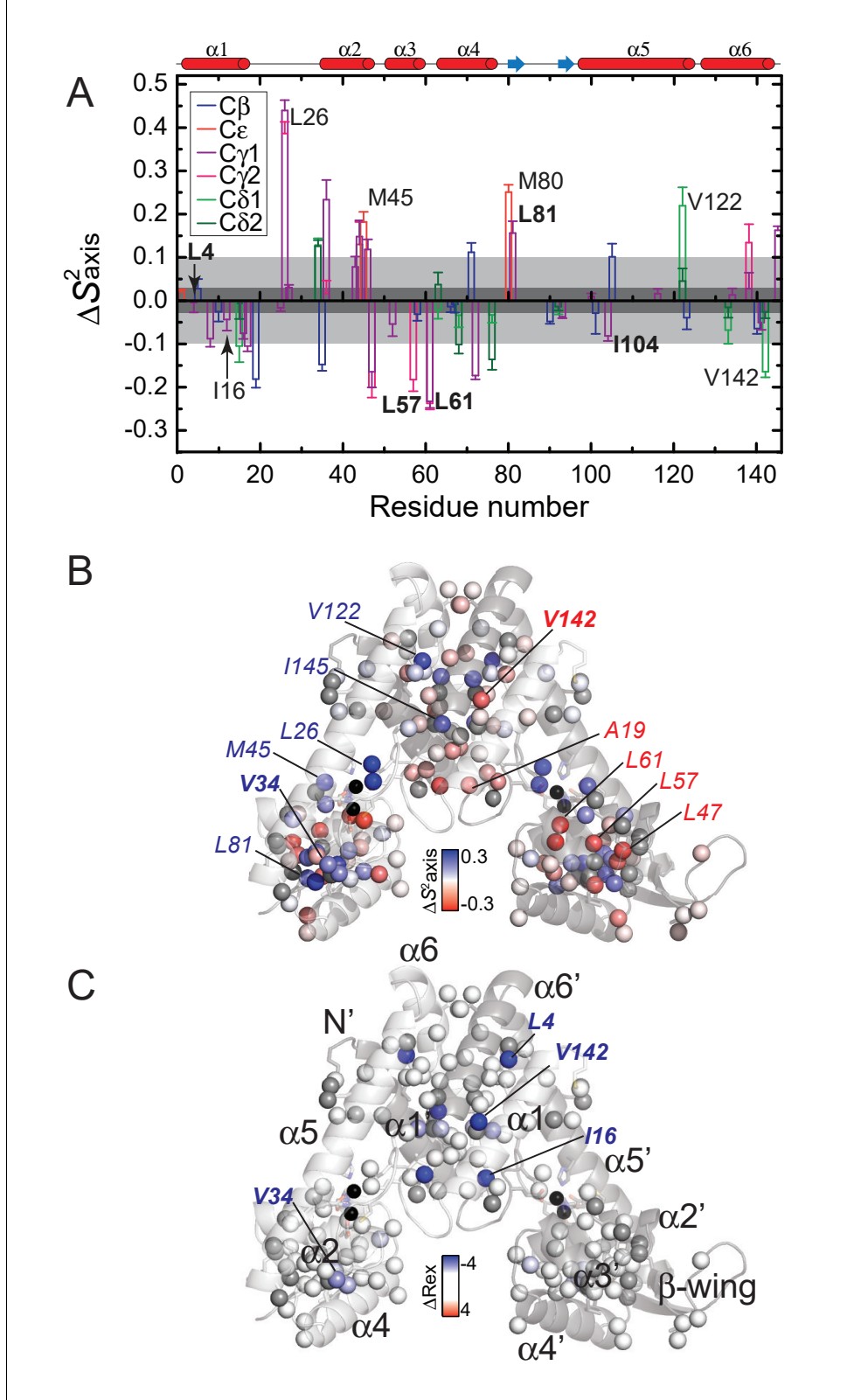

**Figure 5.** Effect of Zn[II] binding to AdcR on the site-specific stereospecifically assigned methyl group axial order parameter. (**A**) Difference in axial order parameter ($\Delta S^2_{axis} = S^2_{axis}{}^{Zn} - S^2_{axis}{}^{apo}$) between apo- and Zn[II]$_2$-states, with the specific type of methyl group color-coded as indicated: C$\beta$, Ala; C$\varepsilon$, Met; C$\gamma$1, C$\gamma$2, Val; C$\delta$1, C$\delta$2, Leu. The dark shaded region shows no significant difference between apo- and Zn-[II]$_2$-bound states and the lighter shaded region represents the cutoff for 'dynamically active' residues. $S^2_{axis}$ (**B**) and $R_{ex}$ (**C**) plotted as $\Delta S^2_{axis}$ ($S^2_{axis}{}^{Zn} - S^2_{axis}{}^{apo}$) and $\Delta R_{ex}$ ($R_{ex}{}^{Zn} - R_{ex}{}^{apo}$) values,

*Figure 5 continued on next page*

*Figure 5 continued*

respectively, mapped onto the structure of $Zn^{II}_2$ AdcR (3tgn). A $\Delta S^2_{axis}$ <0 indicates that the methyl group becomes *more* dynamic in the $Zn^{II}_2$-bound state, while $\Delta R_{ex}$ <0 indicates quenching of motion on the µs-ms timescale in the in the $Zn^{II}_2$-bound state. See *Figure 5—figure supplements 1* and *2* for a graphical representation of all $S^2_{axis}$ and $R_{ex}$ values in each conformation from which these differences were determined, respectively. Residues harboring methyl groups that show major dynamical perturbations on $Zn^{II}$ binding are highlighted, with selected residues subjected to methyl substitution mutagenesis (*Figure 6*).

DOI: https://doi.org/10.7554/eLife.37268.015

The following figure supplements are available for figure 5:

**Figure supplement 1.** Methyl group axial order parameter analysis of AdcR.

DOI: https://doi.org/10.7554/eLife.37268.016

**Figure supplement 2.** Absolute values of methyl group order parameters, $S^2_{axis}$ on the methyl-bearing residues.

DOI: https://doi.org/10.7554/eLife.37268.017

**Figure supplement 3.** Methyl group relaxation dispersion analysis of AdcR.

DOI: https://doi.org/10.7554/eLife.37268.018

and L81 are characterized by significant perturbations in $\Delta S^2_{axis}$ only ($|\Delta S^2_{axis}| \geq 0.2$), with L81 stiffening and L57 and L61 methyls in the α3 helix becoming significantly more dynamic upon $Zn^{II}$ binding (*Figure 5A*, *Table 2*). To investigate the functional role of these residues, we chose missense substitutions (*Table 2*) generally designed to restrict the number of χ angles (Leu to Val or Ala; Val to Ala) and thus impact their dynamical sensitivities (*Capdevila et al., 2017a*; *Capdevila et al., 2018*); in one case (L57), we introduced multiple substitutions, with one characterized by a larger number of χ angles (Leu to Met).

As expected, I104A AdcR is characterized by a DNA binding affinity in the apo- and Zn-states just ≈2-fold lower than wild-type AdcR, returning a $\Delta G_c$ that is not statistically different from wild-type AdcR (*Figure 6C*). Functional characterization of all other methyl substitution mutants in the DNA binding domain results in a ≈5–10-fold decrease or greater (L57V AdcR; *Table 2*) in the DNA binding affinity of the apo-state (*Figure 6C*), with $Zn^{II}$ binding inducing markedly variable degrees of allosteric activation (*Figure 6C*). L36A, closest to the N-terminus of the α2 helix, is most like wild-type AdcR, while L81V AdcR is severely allosterically crippled, with $K_{Zn,DNA}$ some 40-fold lower than wild-type AdcR, and $\Delta G_c$ ≈2-fold lower, from –4.0 to –2.4 kcal mol$^{-1}$. L57M AdcR is even more strongly perturbed ($\Delta G_c \approx$ –2.0 kcal mol$^{-1}$). V34A AdcR shows a comparable degree of functional perturbation, while effectively retaining binding of $Zn^{II}$ only to site 1, like V142A AdcR (discussed below; *Supplementary file 1*-Table S1). We emphasize that these methyl-bearing side chains targeted for substitution are not expected to be in direct contact with the DNA, based on solvent accessible area (*Table 2*) and distance from the DNA binding interface (*Figure 6B*, *Figure 6—figure supplement 5*). With the exception of L36A AdcR, the functional impact of each residue substitution correlates with the magnitude of the dynamical perturbations on that residue. This finding provides additional support for the idea that those methyl-bearing side chains in the DNA-binding domain that exhibit large changes in conformational entropy (as measured by $\Delta S^2_{axis}$) make significant contributions to both DNA binding and allosteric activation by $Zn^{II}$ (*Tzeng and Kalodimos, 2012*; *Capdevila et al., 2017a*). Further characterization of the structural and dynamical impact of these substitutions is necessary to confirm that the functional impact of each is a consequence of dynamical perturbations rather than minor structural changes that would escape detection.

To evaluate the possible contributions of backbone dynamics and structural changes, we purified $^{15}$N-labelled V34A and L57M AdcRs. Unfortunately, the thermal stability of V34A AdcR at the slightly acidic pH and temperature (35°C) required to yield high quality NMR spectra proved inadequate (*Supplementary file 1*-Table S2, *Figure 6—figure supplement 4*) and it was therefore not investigated further. L57M AdcR, on the other hand, yielded excellent quality spectra in both apo and $Zn^{II}_2$ allosteric states, readily yielding backbone resonance assignments (*Figure 6—figure supplement 6*), which could be used to undertake a detailed backbone dynamics characterization. Although the structural changes upon $Zn^{II}$ binding are wild-type-like as reported by a chemical shift perturbation map, the impact of the mutation is not restricted to the α3 helix but also affects the α2 helix as anticipated by the crystal structure (*Figure 6—figure supplement 7*). While the backbone dynamics are largely indistinguishable from wild-type AdcR on both timescales (*Figure 6—figure supplement 8*– *12*), there are several small differences in the DNA-binding domain in the immediate vicinity of M57

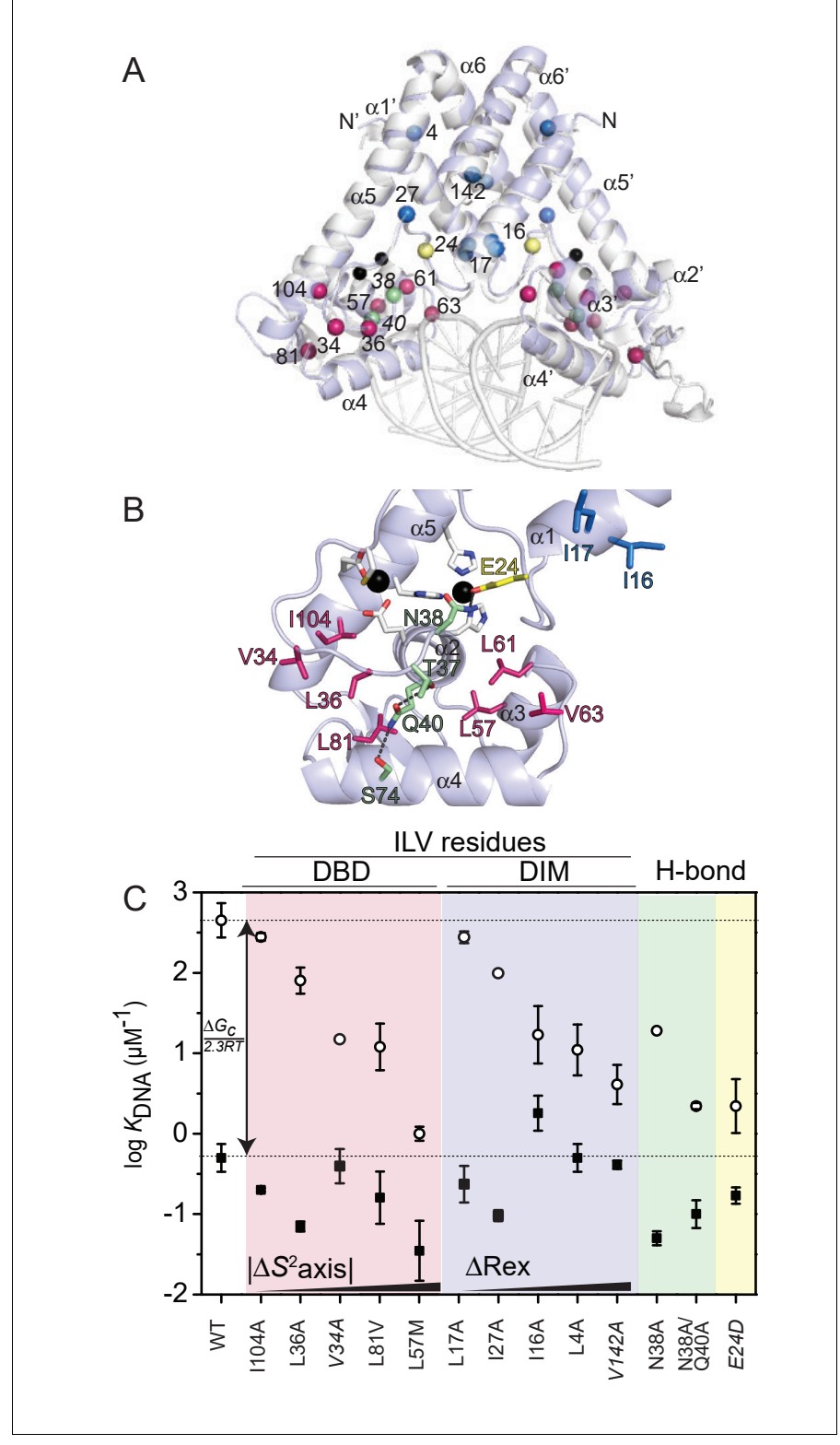

**Figure 6.** Graphical summary of the functional properties of AdcR methyl substitution and hydrogen bonding mutants. (**A**) Cα positions of the residues targeted for methyl substitution mutagenesis in the DNA binding domain (DBD) (*red spheres*) and in the dimerization domain (DIM) (*blue spheres*); other residues targeted for substitution in the hydrogen-bonding pathway (N38, Q40; *green spheres*) and zinc ligand E24 (*yellow spheres*) highlighted on the structure of the Zn$^{II}$$_2$ ZitR-DNA operator complex (**Zhu et al., 2017c**); Zn$^{II}$ ions (*black*

*Figure 6 continued on next page*

*Figure 6 continued*

*spheres*). (**B**) Zoom of the DNA binding domain (DBD) of one of the two $Zn^{II}_2$-bound AdcR protomers highlighting the residues targeted for mutagenesis (methyl substitution mutants, *red stick*; hydrogen-bonding pathway mutants, *green stick*; zinc ligand E24, *yellow stick*), with the helical elements (α1-α5) indicated. (**C**) Coupling free energy analysis for all AdcR mutants highlighted using the same color scheme as in panels A and B. DBD, DNA-binding domain; DIM, dimerization domain; H-bond, hydrogen binding mutants. $K_{DNA}$ for apo-AdcRs are shown in fill circles; $K_{DNA}$ for $Zn^{II}_2$-AdcRs are shown in hollow circles. Lower horizontal line, $K_{DNA}$ for wild-type apo-AdcR; upper horizontal line, $K_{DNA}$ for wild-type $Zn^{II}_2$ AdcR, for reference. The trend in $\Delta S^2_{axis}$ and $\Delta R_{ex}$ is qualitatively indicated (see **Table 2**). These residues are conserved to various degrees in AdcR-like repressors (**Figure 6—figure supplement 5**).

DOI: https://doi.org/10.7554/eLife.37268.019

The following figure supplements are available for figure 6:

**Figure supplement 1.** Representative DNA operator binding isotherms obtained for wild-type (WT) AdcR and selected AdcR mutants in the apo- and $Zn^{II}_2$-states.

DOI: https://doi.org/10.7554/eLife.37268.020

**Figure supplement 2.** Gel filtration chromatograms for AdcR variants in the apo-state.

DOI: https://doi.org/10.7554/eLife.37268.021

**Figure supplement 3.** Representative $Zn^{II}$-binding isotherms obtained from a titration of apo (metal-free) wild-type AdcR or a mutant AdcR and mag-fura-2 (mf2) with $ZnSO_4$.

DOI: https://doi.org/10.7554/eLife.37268.022

**Figure supplement 4.** Differential scanning fluorimetry (DSF) of AdcR and mutants Representative DSF plots acquired with SYPRO orange.

DOI: https://doi.org/10.7554/eLife.37268.023

**Figure supplement 5.** Multiple sequence analysis of AdcRs and closely related MarR family repressors.

DOI: https://doi.org/10.7554/eLife.37268.024

**Figure supplement 6.** Backbone $^1$H-$^{15}$N TROSY spectra showing the resonance assignments of (**A**) apo L57M AdcR and (**B**), $Zn^{II}$ L57M AdcR.

DOI: https://doi.org/10.7554/eLife.37268.025

**Figure supplement 7.** Backbone chemical shift perturbation (CSP) maps for L57M AdcR. Apo-L57M AdcR and for $Zn^{II}_2$ L57M AdcR at pH 5.5, 50 mM NaCl, 35°C relative to WT AdcR as indicated.

DOI: https://doi.org/10.7554/eLife.37268.026

**Figure supplement 8.** $^{15}$N NMR spin relaxation analysis of L57M vs. wild-type AdcR.

DOI: https://doi.org/10.7554/eLife.37268.027

**Figure supplement 9.** Heteronuclear NOE analysis of L57M vs. wild-type AdcR. Apo- and $Zn^{II}_2$ L57M AdcR values of the $^{15}$N-{$^1$H}-NOE (hNOE) are painted onto the 3tgn structure, relative to the WT parameters reproduced here to facilitate comparison.

DOI: https://doi.org/10.7554/eLife.37268.028

**Figure supplement 10.** Backbone dynamics parameters obtained for L57M vs. wild-type AdcRs.

DOI: https://doi.org/10.7554/eLife.37268.029

**Figure supplement 11.** Backbone relaxation dispersion analysis of the L57M vs. wild-type AdcRs.

DOI: https://doi.org/10.7554/eLife.37268.030

**Figure supplement 12.** TROSY NMR spectra of selected AdcR mutants.

DOI: https://doi.org/10.7554/eLife.37268.031

that could contribute to the allosteric impact of the L57M mutation (**Figure 6—figure supplement 8–11**). By and large, however, wild-type and L57M AdcRs are rather dynamically similar along the backbone, thus implicating side chain conformational entropy redistribution as an important contributor to allostery in this system. However, it should be noted that although the structural impact of the L57M mutation is likely small and localized as suggested by the chemical shift perturbation maps (**Figure 6—figure supplement 7**), the effect of a small structural perturbation by M57 can not be ruled out.

## Hydrogen-bonding mutants

A candidate hydrogen-bonding pathway in AdcR (**Chakravorty et al., 2013**) was previously proposed to transmit the $Zn^{II}_2$ binding signal to the DNA binding domain. In this pathway, the Oε1 atom from the $Zn^{II}$ ligand E24 accepts a hydrogen bond from the carboxamide side chain of N38.

**Table 2.** DNA binding parameters for wild-type AdcR and substitution mutants[*]

| AdcR | $K_{apo,DNA}$ (x$10^6$ M$^{-1}$) | $Zn^{II}$ $K_{Zn,DNA}$ (x$10^6$ M$^{-1}$) | $\Delta G_c$ (kcal mol$^{-1}$) | Dynamic changes ($Zn^{II}$) at 600 MHz $\Delta S^2_{axis}$ | $\Delta Rex$ | Fractional ASA[†] |
|---|---|---|---|---|---|---|
| wild-type | 0.5 ± 0.2 | 450 ± 220 | −4.0 ± 0.6 | | | |
| I104A | 0.20 ± 0.01 | 280 ± 30 | −4.3 ± 0.4 | −0.08 ± 0.01 | −0.3 ± 0.6 | 0.04 |
| L36A | 0.07 ± 0.01 | 80 ± 30 | −4.1 ± 0.4 | 0.13 ± 0.10 | −2.0 ± 0.5 | 0.05 |
| V34A | 0.37 ± 0.17 | 13 ± 1 | −2.0 ± 0.3 | 0.13 ± 0.02 | −2.0 ± 0.5 | 0.46 |
| L81V | 0.16 ± 0.12 | 12 ± 8 | −2.4 ± 0.6 | 0.13 ± 0.05 | 0.0 ± 0.5 | 0.00 |
| L61V** | - | - | - | −0.23 ± 0.01 | −1.0 ± 0.5 | 0.01 |
| L57M | 0.035[‡] ± 0.030 | 1 ± 0.2 | −2.0 ± 0.7 | −0.18 ± 0.02 | 1.0 ± 0.5 | 0.00 |
| L57V** | <0.05[§] | <0.05[§] | N/A | −0.18 ± 0.02 | 1.0 ± 0.5 | 0.00 |
| I16A | 1.8 ± 0.9 | 17 ± 14 | −1.8 ± 0.4 | −0.08 ± 0.02 | −4.0 ± 1.0 | 0.11 |
| L4A | 0.5 ± 0.2 | 11 ± 8 | −1.8 ± 0.3 | 0.004 ± 0.045 | −4.0 ± 1.0 | 0.01 |
| V142A | 0.41 ± 0.05 | 4.1 ± 2.3 | −1.4 ± 0.2 | −0.09 ± 0.02 | −3.0 ± 1.0 | 0.31 |
| I27A | 0.09 ± 0.01 | 80 ± 3 | −4.0 ± 0.2 | 0.03 ± 0.01 | 1.2 ± 0.5 | 0.07 |
| L17A | 0.22 ± 0.1 | 219 ± 36 | −4.0 ± 0.2 | −0.10 ± 0.02 | 0.0 ± 0.5 | 0.50 |
| V63A** | - | - | - | 0.01 ± 0.04 | 1.0 ± 0.5 | 0.24 |
| N38A | 0.05 ± 0.01 | 19 ± 10 | −3.5 ± 0.7 | –[#] | – | – |
| N38A/Q40A | 0.10 ± 0.04 | 2.2 ± 0.4 | −1.9 ± 0.2 | – | – | – |
| E24D | 0.17 ± 0.04 | 2.2 ± 1.7 | −1.6 ± 0.3 | – | – | – |

*Conditions: 10 mM Hepes, pH 7.0, 0.23 M NaCl, 1 mM TCEP (chelexed), 10 nM DNA, 25.0°C with 2.0 mM EDTA (for apo-AdcR) or 20 µM ZnCl$_2$ (for $Zn^{II}_2$ AdcR) added to these reactions. See **Figure 6C**, for a graphical representation of these data. All $\Delta G_c$ values lower than −3.5 kcal mol$^{-1}$, with the exception of N38A AdcR are statistically significantly different (p≤0.1) from the wild-type $\Delta G_c$ value.

†Accessible surface area (ASA) was calculated from the $Zn^{II}_2$-bound AdcR (**Guerra et al., 2011**) using the web server for quantitative evaluation of protein structure VADAR 1.8 (vadar.wishartlab.com/)

‡Upper limit on measureable $K_{apo,DNA}$ under these solution conditions.

§ Weaker than upper limit.

#Not measurable using the NMR experiments employed here.

**Significantly lower thermal stability as estimated by differential scanning fluorimetry (**Supplementary file 1**-Table S2) prevented a quantitative analysis of their DNA and metal binding affinities.

DOI: https://doi.org/10.7554/eLife.37268.032

N38 is the +1 residue of the α2 helix, which is then connected to the α4 helix via a hydrogen bond between the Q40 and S74 side chains; further, Q40 accepts a hydrogen bond from the γ-OH of T37 as part of a non-canonical helix N-capping interaction (**Guerra et al., 2011**) (**Figure 6A**). We expect that regardless of the impact that these interactions have on the overall energetics of $Zn^{II}$ binding, they are important in the restriction of fast-time scale dynamics in the α1-α2 loop. We therefore targeted residues E24 (Zn-ligand and H-bond acceptor), N38 and Q40, by characterizing two single mutants, E24D and N38A, and the double mutant, N38A/Q40A AdcR. Although all three mutants undergo conformational switching upon Zn-binding as revealed by $^1$H−$^{15}$N TROSY spectra (**Figure 6—figure supplement 12**) all three exhibit ≈5 − 10-fold decreases in apo-state DNA-binding affinity (**Figure 6C**; **Table 2**). While the single mutant N38A binds $Zn^{II}$ to give $\Delta G_c$ of ≈−3.5 kcal mol$^{-1}$, quite similar to that of wild-type AdcR, in marked contrast, N38A/Q40A AdcR is functionally perturbed, characterized by a $\Delta G_c$ of ≈−1.9 kcal mol$^{-1}$ as is E24D AdcR, which targets a $Zn^{II}$ binding residue (**Figure 6C**). These perturbations provide additional evidence that this hydrogen-bonding pathway may contribute to the motional restriction of the α1-α2 loop, jointly with a redistribution of internal dynamics in the DNA binding domain. This effect can be perturbed directly by mutation of 'dynamically active' sidechains (L81V, L57M) or by significantly impacting the interactions that restrict the loop (N38A/Q40A).

## Dimerization domain mutants

To test the functional role of the dimerization domain in dynamical changes, we targeted four methyl-bearing residues in this domain, including L4, I16 and L17 on opposite ends of the α1 helix; V142, near the C-terminus of the α6 helix (*Figure 6B*) and I27 a α1-α2 loop in the proximity of V142. I16 and L17 are closer to the intervening minor groove of the DNA operator, while V142, I27 and L4 are increasingly distant from the DNA. With the exception of L17 and I27, these side chains are primarily active in slow timescale dynamics, with $Zn^{II}$-binding quenching side chain mobility on the µs-ms timescale, that is, global motions, but relatively smaller changes in $\Delta S^2_{axis}$ (*Figure 5B*; *Table 2*). Methyl substitution mutants of these residues (I16A, L4A and V142A) bind DNA in the apo-state with wild-type like affinities, but each is allosterically strongly perturbed, with only $\approx 10 - 20$-fold allosteric activation by $Zn^{II}$, giving $\Delta G_c$ values of –1.4 to –1.8 kcal mol$^{-1}$. On the contrary, L17A and I27A AdcR shows a wild-type-like $\Delta G_c$, consistent with the fact that L17 and I27 are nearly dynamically silent upon Zn binding (*Figure 5B*).

These findings suggest that $Zn^{II}$-dependent quenching of global motions far from the DNA binding domain play a significant role in allostery in this system. Our characterization of allosterically compromised mutants that affect site-specific conformational entropy (L81V, L57M) and conformational exchange (V34A, L4A, I16A) provides evidence for two classes of functional dynamics in AdcR that comprise different regions of the molecule, operating on different timescales (from sub-nanoseconds to milliseconds). Thus, we propose that a $Zn^{II}$-dependent redistribution of internal dynamics quenches global, slow and fast motions in the dimer, yet detectably enhances local dynamical disorder in the DNA binding domain, which we propose can ultimately be harnessed to maximize contacts at the protein-DNA interface.

## Conclusions

Members of the multiple antibiotic resistance repressor (MarR) family of proteins comprise at least 12,000 members (*Capdevila et al., 2017b*), and many have been subjected to significant structural inquiry since the original discovery of the *E. coli mar* operon and characterization of *E. coli* MarR some 25 years ago (*Cohen et al., 1993*; *Seoane and Levy, 1995*). The crystallographic structure of this prototypical *E. coli* MarR appeared a few years later (*Alekshun et al., 2001*) and has inspired considerable efforts to understand the inducer specificity and mechanisms of transcriptional regulation in *E. coli* MarR (*Hao et al., 2014*) and other MarR family repressors (*Grove, 2013*), which collectively respond to an wide range of stimuli, including small molecules, metal ions, antibiotics and oxidative stress (*Deochand and Grove, 2017*). We have examined the wealth of crystallographic data available from 135 MarR family repressor structures solved in a variety of functional states, including DNA-binding competent, DNA-binding incompetent and DNA-bound states (*Figure 1*). This analysis of the crystal structures suggests that a conformational ensemble model of allostery must be operative in a significant number of these repressor systems, where ligand binding or thiol oxidation narrows the conformational spread and, thus, activates or inhibits DNA binding. Here, we present the first site-specific dynamics analysis of any MarR family repressor in solution, and establish that conformational dynamics on a range of timescales is a central feature of $Zn^{II}$-dependent allosteric activation of DNA operator binding by the zinc uptake regulator *S. pneumoniae* AdcR (*Reyes-Caballero et al., 2010*) and closely related repressors (*Zhu et al., 2017c*).

We explored dynamics in the sub-nanosecond and ms timescales with residue-specific resolution, both along the backbone, as measured by N-H bond vectors, and in the methyl groups of the methyl-bearing side chains of Ala, Met, Val, Leu and Ile. These measurements, coupled with small angle x-ray scattering measurements of both conformational states, lead to a self-consistent picture of allosteric activation by $Zn^{II}$ in AdcR. The apo-state conformational ensemble is far broader than the $Zn^{II}_2$ state, and features at least partial dynamical uncoupling of the core DNA-binding and dimerization domains, facilitated by rapid motions in the α1-α2 loop and the α5 helix in the immediate vicinity of the $Zn^{II}$ coordinating residues. This motion is superimposed on much slower motions across the dimerization domain, far from the DNA interface, which affect both backbone amide and side chain methyl groups (*Figures 4–5*). $Zn^{II}$ binding substantially quenches both the low amplitude internal motions and global, larger amplitude movements like the ones reflected by SAXS data, with an accompanying redistribution of these dynamics into the DNA-binding domain.

As observed previously for another $Zn^{II}$ metalloregulatory protein (*Capdevila et al., 2018*), $Zn^{II}$ binding induces a small, net global conformational stiffening of the internal dynamics or sub-ns motions; however, in AdcR, there are significant contributions from *both* the backbone (in folding the α1-α2 loop) and the methyl-bearing side chains upon $Zn^{II}$ binding. These are superimposed on pockets of increased dynamical disorder, particularly in the α2-α3 loop along the backbone (*Figure 4*), and in the α3-α4 region of the DNA binding domain (*Figure 5*). To test the functional importance of both these fast-time scale motions in the DNA binding domain, as well as slow timescale dynamics in the dimerization domain, we exploited these side chain dynamics results (*Figure 5*) (*Capdevila et al., 2017a*) to guide our introduction of methyl substitutions of both dynamically active and dynamically silent residues (*Figure 6*). We generally find that methyl substitutions in the DNA binding domain are strongly deleterious for residues that are dynamically active in the fast timescale ($|\Delta S^2_{axis}|{>}0.2$), that is L81, L61, L57. The same is true of dynamically active slow timescale residues,that is L4, I16 and V142. These findings confirm a functional role of these pronounced changes in dynamics (*Capdevila et al., 2017a*; *Capdevila et al., 2018*) and suggest that $Zn^{II}_2$-bound AdcR has an optimal distribution of internal millisecond dynamics that if perturbed, leads to weakened DNA binding affinity in the allosterically active Zn-bound state.

The extent to which this dynamics-centered regulatory model characterizes other MarR family repressors in solution is of course unknown. However, the differences between the crystal structures of the DNA binding-competent and incompetent states appear sufficient to adequately describe the allosteric mechanism in only a handful of MarR repressors (*Figure 1*). From this perspective, it is interesting to speculate on the evolutionary origin of allosteric activation and allosteric inhibition within this simple molecular scaffold. Clearly, models that invoke only rigid body domain motions as contributing to allostery (*Alekshun et al., 2001*; *Chang et al., 2010*; *Dolan et al., 2011*; *Saridakis et al., 2008*; *Birukou et al., 2014*; *Radhakrishnan et al., 2014*) would fail to capture the evolution of allosteric activation vs. inhibition from a common progenitor repressor (*Motlagh et al., 2014*). Further, we have previously speculated that nature is capable of harnessing dynamics properties and entropy reservoirs to evolve new inducer specificities in another structural class of bacterial repressors (*Capdevila et al., 2017a*).

Here, we propose that both internal dynamics, reflected in a more favorable conformational entropy term, and structural features, reflected in a more favorable $\Delta H$ term, were originally optimized in a common progenitor MarR that was capable of transcriptionally repressing genes that became deleterious when colonizing a new environment (*Deochand and Grove, 2017*). Then, any set of sequence variations could allow for the emergence of both allosteric activation and inhibition. For example, introduction of a dynamic element(s), that is loops or disordered regions (*Pabis et al., 2018*; *Campbell et al., 2016*) would impact both coupled fast sub-ns motions and concerted slower motions and as a result, introduce an entropic penalty that leads to inhibition of DNA-binding. Indeed, a structural comparison and an extensive multiple sequence alignment reveals that only AdcR-like repressors harbor an α1-α2 loop larger than 10 residues (*Figure 7A*), and that ligand ($Zn^{II}$) binding to what we now know is a highly dynamical loop element, becomes an important feature of allosteric activation of DNA binding.

On the other hand, allosteric inhibition could have arisen from sequence variations that define a pocket where ligand binding disrupts structural (*Hong et al., 2005*; *Dolan et al., 2011*; *Quade et al., 2012*; *Birukou et al., 2014*; *Zhu et al., 2017a*; *Gao et al., 2017*; *Otani et al., 2016*) and/or dynamical features (*Capdevila et al., 2017a*) of a DNA binding-competent conformation (*Figure 7B*). Although the presence of functionally important entropic reservoirs on any allosterically inhibited MarR has not yet been reported experimentally, molecular dynamics simulations show that DNA binding-impaired mutants of MexR differ from the wild-type repressor in the nature of the dynamical connection between the dimerization and DNA binding domains (*Anandapadamanaban et al., 2016*). This dynamical connectivity is in fact exploited by the binding the ArmR peptide, leading to DNA dissociation (*Anandapadamanaban et al., 2016*; *Wilke et al., 2008*). We propose that conformational entropy can contribute to other mechanisms of allosteric inhibition to yield a repressor that binds tightly to the operator sequence and yet has the ability to readily evolve new inducer specificities.

It is interesting to note that mutations that lead to inactivation are not necessarily part of a physical pathway with the DNA binding site (*Clarke et al., 2016*), since they only need to affect dynamical properties that are likely delocalized in an extended network. Notably, single point mutants in the

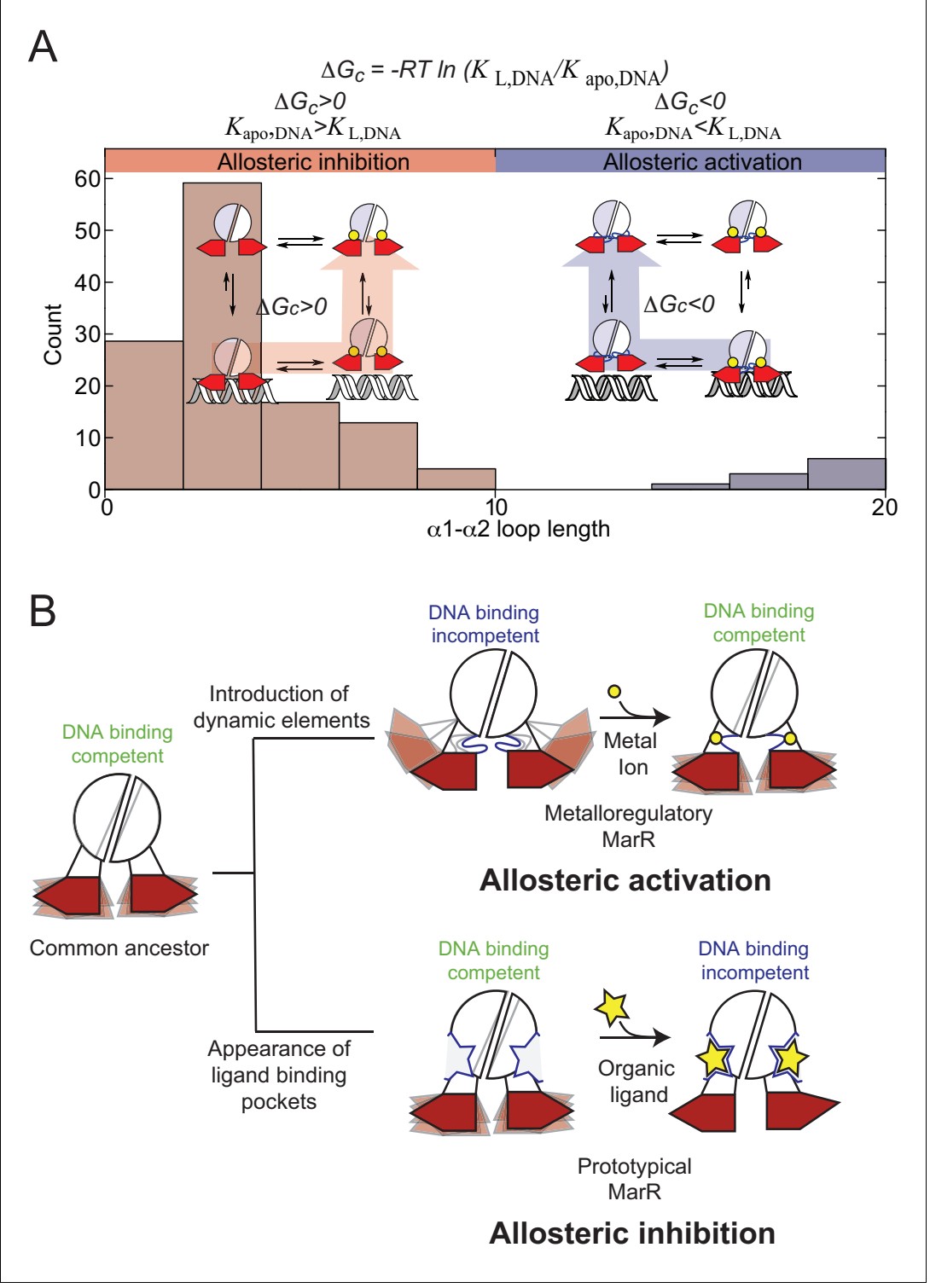

**Figure 7.** (A) Histogram representation of the distribution of α1-α2 loops lengths in the reported structures in MarR family of proteins, with the bars are colored to account for the measured or proposed coupling free energies in each case. Proteins that are DNA binding-competent in the apo- state and DNA binding-incompetent in the ligand-bound state are colored in *red*, while proteins that are DNA binding-incompetent in the apo-state and DNA binding-competent in the liganded state are colored in *blue* (see *Figure 1—source data 1* for a full accounting of these structures). A schematic representation of allosteric inhibition and activation are shown (*inset*), with shorter α1-α2 loops associated with allosteric inhibition of DNA binding upon ligand binding, while longer

*Figure 7 continued on next page*

*Figure 7 continued*

loops are associated with allosteric activation (like that for AdcR/ZitR) upon ligand binding. (B) Dynamically driven model for how conformational dynamics can be harnessed to evolve allosteric activation (*upper right*) vs. allosteric inhibition (*lower right*) in the same molecular scaffold. This model suggests that dynamic properties of the DNA binding competent states have been conserved to give rise to a more favorable conformational entropy. In the metalloregulatory MarRs (AdcR, ZitR), the inactive state shows perturbed dynamics over a range of timescales; apo-AdcR therefore exhibits low affinity for DNA. Metal ion (*yellow circle*) coordination quenches both local and global modes in the dimerization domain and linkers, while inducing conformational disorder in the DNA-binding domain that enhances DNA binding affinity, thus stabilizing a conformation that has high affinity for DNA and giving rise to a favorable conformational entropy. For prototypical MarRs, where the ligand (*yellow star*) is an allosteric inhibitor, ligand binding narrows the conformational ensemble to a DNA-binding incompetent conformation decreasing the enthalpic contribution to DNA binding.
DOI: https://doi.org/10.7554/eLife.37268.033

dimerization domain of various MarR family repressors have been shown to modulate allostery and DNA binding (*Anandapadamanaban et al., 2016*; *Deochand et al., 2016*; *Liguori et al., 2016*; *Duval et al., 2013*; *Alekshun and Levy, 1999*; *Andrésen et al., 2010*), perhaps exemplified by the L4, I16 and V142 AdcR substitution mutants. In AdcR, while structural perturbations induced by $Zn^{II}$ binding are essentially confined to the $Zn^{II}$ binding pocket, dynamical perturbations extend all over the molecule, and feature many residues that are far from either ligand binding site, and are dynamically active on the sub-nanosecond and/or μs-ms timescales (*Figures 4–5*). Thus, a conformational entropy contribution that is inherently delocalized and easily perturbed can enable rapid optimization of new inactivation mechanisms that would allow new biological functionalities to emerge (*Figure 7*). These findings inspire efforts to explore the evolution of allostery in this remarkable family of transcriptional repressors, by exploiting an allosterically crippled AdcR, for example L57M AdcR, to re-evolve allostery in this system.

# Materials and methods

## AdcR mutant plasmid production
An overexpression plasmid for *S. pneumoniae* AdcR in a pET3a vector was obtained as previously described and was used as a template for the production of all mutant plasmids (*Reyes-Caballero et al., 2010*). Mutant AdcR plasmids were constructed by PCR-based site-directed mutagenesis, and verified using DNA sequencing.

## Protein production and purification
AdcR plasmids were transformed into either *E. coli* BL21(DE3) pLysS or Rosetta cells. *E. coli* cultures were either grown in LB media or M9 minimal media supplemented with $^{15}NH_4Cl$ as the sole nitrogen source with simple $^1H,^{15}N$ HSQC spectroscopy to assess the structural integrity of selected mutant proteins. Protein samples for backbone and methyl group assignments of AdcR were isotopically labeled using published procedures as described in our previous work (*Capdevila et al., 2017a*; *Arunkumar et al., 2007*), with all isotopes for NMR experiments purchased from Cambridge Isotope Laboratories. Protein expression and purification were carried out essentially as previously described (*Reyes-Caballero et al., 2010*). All proteins were confirmed to have <0.05 molar equivalents of Zn(II) as measured by atomic absorption spectroscopy and were dimeric by gel filtration chromatography. The AdcR protein concentration was measured using the estimated molar extinction coefficient at 280 nm of 2980 $M^{-1}$ $cm^{-1}$.

## Small angle x-ray scattering experiments
Small angle and wide angle x-ray scattering data of the apo and $Zn^{II}_2$ states of AdcR was collected at three different protein concentrations (5 mg/mL, 2.5 mg/mL and 1.25 mg/mL) in buffer 25 mM MES pH 5.5, 400 mM NaCl, 2 mM EDTA/10 μM $ZnCl_2$, 2 mM TCEP at sector 12ID-B at the Advanced Photo Source (APS) at Argonne National Laboratory. For each protein concentration and matching background buffer, 30 images were collected and averaged using NCI-SAXS program package. The

scattering profile at each concentration was manually adjusted with the scale factor to remove the effect of concentration prior to subtraction of the scattering profile of the buffer. Scattering profiles of each protein concentration were then merged for further analysis. The GUINIER region was plotted with ln (I($q$)) vs $q^2$ to check for monodispersity of the sample and to obtain $I_0$ and the radius of gyration ($R_g$) within the range of $q_{max}*R_g$ <1.3. The $R_g$ values obtained for apo-AdcR and Zn(II)-bound-AdcR are 25.5 ± 0.9 Å and 23.7 ± 1.1 Å, respectively. The scattering profiles of each AdcR conformational state was then normalized with $I_0$. The compaction of each states of AdcR was examined using the Kratky plot for $q < 0.3$ Å$^{-1}$. Scattering profiles for apo and Zn$^{II}_2$ states of AdcR were then Fourier-transformed using GNOM of the ATSAS package to obtain the normalized pair-wise distance distribution graph (PDDF).

*Ab initio* modeling was performed using the program DAMMIF in a slow mode (*Franke and Svergun, 2009*). For each conformational state of AdcR, 10 models were obtained. These models were compared, aligned and averaged using the DAMSEL, DAMSUP, DAMAVER, DAMFILT, respectively, as described in the ATSAS package (http://www.embl-hamburg.de/bioSAXS). Normalized spatial discrepancy (NSD) between each pair of the models was computed. The model with the lowest NSD value was selected as the reference against which the other models were superimposed. Outliner models (two models) with an NSD above mean +2*standard deviation of NSD were removed before averaging. For refinement, the averaged envelope of the first run was used as search volume for the second round of modeling. Modeling of the envelope of apo-AdcR was restrained by enforcing $P_2$ rotational symmetry while that Zn$^{II}_2$ AdcR was restrained using compact, hallow and no-penalty constraints. Scattering profiles of crystal structures were calculated using the fast x-ray scattering (FOXS) webserver (https://modbase.compbio.ucsf.edu/foxs/) (*Schneidman-Duhovny et al., 2010*).

## NMR spectroscopy

NMR spectra were acquired on a Varian VNMRS 600 or 800 MHz spectrometer, each equipped with a cryogenic probe, at the Indiana University METACyt Biomolecular NMR laboratory. The two-dimensional spectra were processed using NMRPipe (*Delaglio et al., 1995*). The three-dimensional spectra were acquired using Poisson-gap non-uniform sampling and reconstructed using hmsIST (*Hyberts et al., 2012*) and analyzed using Sparky (*Lee et al., 2015*) or CARA (http://cara.nmr.ch). Typical solution conditions were ~500 µM protein (protomer), 25 mM MES pH 5.5, 50 mM NaCl, 1 mM TCEP, 0.02% (w/v) NaN$_3$, and 10% D$_2$O. Some spectra were recorded at pH 6.0 as indicated. Our previous NMR studies of AdcR (*Guerra et al., 2011*; *Guerra and Giedroc, 2014*) were carried out with samples containing ≈70% random fractional deuteration, pH 6.0, 50 mM NaCl, 35°C; under those conditions, the backbone amides of residues 21 – 26 in the α1-α2 loop and harboring zinc ligand E24 as well as the N-terminal region of the α2 helix (residues 37 – 40) exhibited significant conformational exchange broadening in the apo-state and could not be assigned (*Guerra et al., 2011*). In this work, we acquired comprehensive $^1$H-$^{15}$N TROSY-edited NMR data sets at 600 and 800 MHz for a 100% deuterated AdcR sample in both apo- and Zn$_2$-bound states at pH 5.5, 50 mM NaCl, 35° C. Under these conditions, only four backbone amides residues in the apo-state were broadened beyond detection (residues 21, 38 – 40); all were visible and therefore assignable in the Zn$^{II}_2$ state. Thus, the N-terminus of the α2 helix, including N38 and Q40 are clearly exchange broadened in the apo-state. Sidechains were assigned following published procedures as described in our previous work (*Capdevila et al., 2017a*; *Arunkumar et al., 2007*). The Leu and Val methyl resonances were distinguished using through-bond information such as HMCMCBCA or HMCM[CG]CBCA experiments (*Tugarinov and Kay, 2003*) which correlate the Leu or Val methyl resonances with other side chain carbon resonances. All apo-protein samples contained 1 mM EDTA. All Zn$^{II}_2$ samples contained two monomer mol equiv of Zn$^{II}$. Chemical shifts were referenced to 2,2-dimethyl-2-silapentane-5-sulfonic acid (DSS; Sigma) (*Wishart and Sykes, 1994*). Chemical shift perturbations (CSP) of the backbone and methyl groups upon Zn$^{II}$ binding or mutation were calculated using $^1$H and $^{15}$N chemical shifts of the methyl groups ($\Delta\delta=(\Delta\delta_H)^2+ 0.2(\Delta\delta_N)^2$) and $^1$H and $^{13}$C chemical shifts of the methyl groups ($\Delta\delta=(\Delta\delta_H)^2+ 0.3(\Delta\delta_C)^2$), respectively.

$^{15}$N spin relaxation rates, $R_1$ and $R_2$, and $^1$H-$^{15}$N heteronuclear NOE (hNOE) values were measured using TROSY pulse sequences described elsewhere (*Zhu et al., 2000*) on the 100% deuterated AdcR sample. The relaxation delays used were 0.01, 0.05, 0.11, 0.19, 0.31, 0.65, 1, 1.5, 1.9, 2.3, 2.7, and 3.2 s for $R_1$ and 0.01, 0.03, 0.05, 0.07, 0.09, 0.11, 0.13, 0.15, 0.19, and 0.25 s for $R_2$. Residue-specific $R_1$ and $R_2$ values were obtained from fits of peak intensities vs. relaxation time to a single

exponential decay function, while hNOE ratios were ascertained directly from intensities in experiments recorded with (2 s relaxation delay followed by 3 s saturation) and without saturation (relaxation delay of 5 s). Theoretical hNOEs values were estimated using the Solomon equation that takes into account the fact that the recycle delay is not much longer than $T_1$ (*Gong and Ishima, 2007*; *Freedberg et al., 2002*; *Lakomek et al., 2012*). Errors in hNOE values were calculated by propagating the error from the signal to noise.

Values of rotational correlation times were obtained from Monte Carlo simulations with tensor2 software (*Dosset et al., 2000*), using $T_1$, $T_2$, and heteronuclear NOE (hNOE) recorded at 35°C at 800 MHz, in 10% $D_2O$ (*Figure 4—figure supplement 2*). A chemical shift anisotropy (CSA) angle of value of 17 degrees was used for these calculations. For apo- and $Zn^{II}_2$ AdcRs, the $\tau_c$ obtained in this way is 16.9 ± 0.1 ns and 21.1 ± 0.1 ns respectively. The results for $Zn^{II}_2$-AdcR were in very good agreement with the correlation time and relaxation rates obtained from HydroNMR (*García de la Torre et al., 2000*) for the crystal structure of $Zn^{II}_2$-AdcR (3tgn, $\tau_c$=20 ns, *Figure 4—figure supplement 1*, *grey* lines). A value of the atomic radius element of 3.2 Å and the known viscosity for water at 35°C (*Cho et al., 1999*) were used for this calculation.

$S^2_{axis}$ of the Ile δ1, Leu δ1/δ2, Val γ1/γ2, Ala β, and Met ε methyl groups in apo and Zn(II)$_2$ states were determined using $^1H$ spin-based relaxation experiments at 600 MHz at 35.0°C (*Tugarinov et al., 2007*). $S^2_{axis}$ values, cross-correlated relaxation rates, η, between pairs of $^1H$–$^1H$ vectors in $^{13}CH_3$ methyl groups were measured using *Equation. 2*

$$\eta = \frac{R^F_{2,H} - R^S_{2,H}}{2} \approx \frac{9}{10}\left(\frac{\mu_o}{4\pi}\right)^2\left[P_2\left(cos\theta_{axis,HH}\right)\right]^2\frac{2S^2_{axis}\gamma^4_H\hbar^2\tau_c}{r^6_{HH}} \tag{2}$$

where $\tau_c$ is the tumbling time of the protein; $R^F_{2,H}$ and $R^S_{2,H}$ are the fast and slow relaxing magnetization, respectively; $\gamma_H$ is the gyromagnetic ratio of the proton; and $r_{HH}$ is the distance between pairs of methyl protons.

In order to obtain an approximation of the differences in fast and slow relaxation rates (2η, we measured the time-dependence of the cross peak intensities in a correlated pair of single and double quantum (2Q) experiments (*Tugarinov et al., 2007*). Using various delay time, *T*, values (3, 5, 8, 12, 17, 22, and 27 ms, recorded in an interleaved manner), the rates of η were obtained by fitting ratios of peak intensities measured in pairs of experiments ($I_a$ and $I_b$, spin-forbidden and spin-allowed, respectively) with *Equation. 3*:

$$\frac{I_a}{I_b} = \frac{-0.5\eta\tanh\left(\sqrt{\eta^2 + \delta^2}T\right)}{\sqrt{\eta^2 + \delta^2} - \delta\tanh\left(\sqrt{\eta^2 + \delta^2}T\right)} \tag{3}$$

where *T* is the variable delay time, δ is a parameter that is related to the $^1H$ spin density around the methyl group, and $I_a$ and $I_b$ are the time dependencies of differences and sums, respectively, of magnetization derived from methyl $^1H$ single-quantum transitions, as described (*Tugarinov et al., 2007*). Peak heights and spectral noise were measured in Sparky (*Lee et al., 2015*). A python script (*Source code 1*) was used to fit the peak height ratios to η values and to determine $S^2_{axis}$ values in the apo- or Zn-bound states, as described previously (*Tugarinov and Kay, 2004*; *Tugarinov et al., 2007*; *Capdevila et al., 2017a*). $\tau_c$ was obtained from Monte Carlo simulations with tensor2 software.

The conformational entropy between Zn and apo states was obtained using a methyl order parameters, $S^2_{axis}$, as dynamical proxy (*Caro et al., 2017*):

$$-TS_{CONF,sc,a\rightarrow b} = -T\left(-0.00116\,\mathrm{kcalmol^{-1}K^{-1}}\right)N^{prot}_\chi\left[\langle S^2_b\rangle - \langle S^2_a\rangle\right] \tag{4}$$

where $N^{prot}_\chi$ is the total number of side-chain torsion angles in the protein dimer.

We also evaluated the contribution of the changes in the backbone dynamics using previously reported calibration curve for backbone entropy obtained from molecular dynamics simulations (*Sharp et al., 2015*):

$$-T\Delta S_{conf,bb,a\rightarrow b} = -T\left(0.0017\,\mathrm{kcalmol^{-1}K^{-1}}\right)N^{prot}_{res}\left[\left\langle ln\left(1 - S^2_{NH,b}\right) - ln\left(1 - S^2_{NH,a}\right)\right\rangle\right] \tag{5}$$

where $N_{res}^{prot}$ is the total number of residues in the protein dimer (292 in the case of AdcR). This calculation was performed only for residues that had $S_{NH}^2 < 0.8$ in at least one of the allosteric states.

Relaxation dispersion measurements were acquired using a TROSY adaptation of $^{15}$N and a $^1$H-$^{13}$C HMQC-based Carr–Purcell–Meiboom–Gill (CPMG) pulse sequence for amides from the backbone (*Tollinger et al., 2001*) and methyl groups from the sidechains (*Korzhnev et al., 2004*), respectively. Experiments were performed at 35°C at 600 and 800 MHz $^1$H frequencies using constant time interval $T = 40$ ms with CPMG field strengths ($\nu_{CPMG}$) of 50, 100, 150, 200, 250, 300, 350, 400, 450, 500, 600, 700, 850, and 1,000 Hz. Peak intensities in CPMG experiments were converted to effective transverse relaxation rates ($R_{2,eff}$) using the equation, $R_{2,eff} = (-1/T) \ln(I/I_0)$, where I and $I_0$ are peak intensities measured with and without the CPMG delay (*Korzhnev et al., 2004*). We estimated the exchange regime from the analysis of the $R_{2,eff}$ dependence with the $B_0$ (*Millet et al., 2000*). Since all the measured probes had values compatible with a fast exchange regime, variation in $R_{2,eff}$ as a function of CPMG pulsing frequency was fit to:

$$R_{2,\,eff} = R_2 + R_{ex}.\left[1 - 2\tau\,\nu_{CPMG}\,\tanh\left(\frac{1}{2\tau.\nu_{CPMG}}\right)\right] \tag{6}$$

The authors note that this analysis fails to provide several additional details that could be obtained from the full Carver-Richards equations such as populations and chemical shift differences, however to obtain those parameters it is necessary to have a significant number of probes in slow or intermediate exchange (*Kovrigin et al., 2006*). Most of the probes that show significant exchange share similar values of $\tau$ and there was no significant improvement in the fit using a residue-specific $\tau$, so a two-state model was preferred (*Source code 2*). The global $\tau$ for each state was obtained by averaging the fitted $\tau$s for all well-fit probes showing significant exchange, and evaluated by the reduced $\chi 2$ (*Source code 3*). $R_{ex}$ values were included in the analysis only if the reduced $\chi 2$ value for the fit fell under the threshold of 1.7. The $\chi 2$ values for representative probes are shown in *Figure 4—figure supplement 3*, *Figure 5—figure supplement 3*, and *Figure 6—figure supplement 11*.

## DNA binding experiments and determination of allosteric coupling free energies ($\Delta G_c$)

For all DNA binding experiments a 28 bp double stranded DNA was obtained as previously described (*Reyes-Caballero et al., 2010*) with the following sequence of the AdcO: 5'-TGATATAAT-TAACTGGTAAACAAAATGT[F]−3'. Apo AdcR binding experiments were conducted in solution conditions of 10 mM HEPES, pH 7.0, 0.23 M NaCl, 1 mM TCEP (chelexed), 10 nM DNA, 25.0°C with 2.0 mM EDTA (for apo-AdcR) or 20 µM $ZnCl_2$ (for $Zn^{II}_2$ AdcR) added to these reactions. Anisotropy experiments were performed on an ISS PC1 spectrofluorometer in steady-state mode with Glan-Thompson polarizers in the L-format. The excitation wavelength was set at 494 nm with a 1 mm slit and the total emission intensity collected through a 515 nm filter. For Zn(II)-bound-AdcR DNA-binding experiments, the data were fit with DynaFit (*Kuzmic, 1996*) using a non-dissociable dimer 1:1 dimer:DNA binding model ($K_{dim} = 10^{12}$ M$^{-1}$) (*Source code 4*). For Zn(II)-bound experiments, the initial anisotropy ($r_0$) was fixed to the measured value for the free DNA, with the anisotropy response of the saturated protein:DNA complex ($r_{complex}$) optimized during a nonlinear least squares fit using DynaFit (*Kuzmic, 1996*). Apo binding data were fit in the same manner, except $r_{complex}$ was fixed to reflect the anisotropy change ($r_{complex} - r_0$) observed for wild-type AdcR in the presence of zinc. The errors on $K_{apo,DNA}$ and $K_{Zn,DNA,}$ reflect the standard deviation of 3 independent titrations (*Table 2*). The coupling free energies were calculated using the following equation:

$\Delta G_c = -RT\ln(K_{Zn,DNA}/K_{apo,DNA})$ (*Giedroc and Arunkumar, 2007*). Negative values of $\Delta G_c$ were observed since AdcR is a positive allosteric activator in the presence of $Zn^{II}$ ($K_{apo,DNA} < K_{Zn,DNA,}$).

## Mag-fura-2 competition assays

All mag-fura-2 competition experiments were performed on an ISS PC1 spectrofluorometer in operating steady-state mode or a HP8453 UV-Vis spectrophotometer as described in our previous work (*Capdevila et al., 2017a*; *Campanello et al., 2013*) using the following solution conditions: 10 mM Hepes, pH 7.2, 400 mM NaCl that was Chelex (Bio-rad) treated to remove contaminating metals. 10 mM protein concentration was used for all and MF2 concentration ranged from 13 to 16 µM. These

data were fit using a competitive binding model with DynaFit (*Kuzmic, 1996*) (*Source code 5*) to determine zinc binding affinities for wild-type and each mutant AdcR using a four-site-nondissociable homodimer binding model, as previously described (*Reyes-Caballero et al., 2010*) with $K_{Zn} = 4.9 \times 10^6$ M$^{-1}$ for mag-fura-2 fixed in these fits. $K_1$ and $K_2$ correspond to filling the two high affinity sites (site 1), and only a lower limits ($\geq 10^9$ M$^{-1}$) could be obtained for these sites; $K_3$ and $K_4$ were allowed to vary in the fit, and are reported in *Supplementary file 1*-Table S1. Experiments were conducted three times for each AdcR variant. Errors of the binding constant parameters were estimated from global fits.

## SYPRO orange Differential Scanning Fluorimetry assays

All SYPRO Orange assays were done in triplicate 25 µL reactions on a 96-well plate in a PCR machine in a chelexed buffer containing 10 mM Hepes, pH 7.0, 0.23 M NaCl, 1 mM TCEP. 4 – 8 µM protein concentration and 5x SYPRO orange were added to all reactions (*Niesen et al., 2007*). 10 µM EDTA was added to apo-AdcR melts to remove any contaminating metals from apo-AdcR samples. For Zn$^{II}_2$ AdcR samples, two protomer mol-equivalents of ZnCl$_2$ were added to these reactions (for Zn$^{II}_2$ AdcR). Other assays were carried out in solution conditions used for NMR spectroscopy, 25 mM MES, pH 5.5, 50 mM NaCl, 1 mM TCEP (chelexed), and 4 – 8 µM protein concentration and 5x SYPRO orange. The temperature was increased from 25°C to 95°C at a ramp rate of 1°C per minute. Apparent melting temperatures ($T_m$) were determined from the maximum of the first derivative of the florescence signal in each data set. Errors were determined from the standard deviation derived from triplicate measurements.

## Acknowledgements

We gratefully acknowledge support of this work by the NIH (R35 GM118157 to DPG). NMR instrumentation in the METACyt Biomolecular NMR Laboratory at Indiana University was generously supported by a grant from the Lilly Endowment. DAC acknowledges support from the Pew Latin American Fellows Program in the Biomedical Sciences and the *Fundación William*s. We also thank Dr. Lixin Fan of the Small-Angle X-ray Scattering Core Facility, National Cancer Institute, Frederick, MD for acquiring the SAXS data. We appreciate the use of pulse sequences modified by Dr. Marco Tonelli, NMRFAM, for $T_1$, $T_2$, and heteronuclear NMR (hNOE) measurements.

## Additional information

### Funding

| Funder | Grant reference number | Author |
| --- | --- | --- |
| NIH Office of the Director | GM118157 | David P Giedroc |
| Pew Charitable Trusts | Latin American Fellowship | Daiana A Capdevila |

The funders had no role in study design, data collection and interpretation, or the decision to submit the work for publication.

### Author contributions

Daiana A Capdevila, Conceptualization, Formal analysis, Supervision, Investigation, Writing—original draft; Fidel Huerta, My Tra Le, Formal analysis, Investigation; Katherine A Edmonds, Data curation, Formal analysis, Investigation, Writing—review and editing; Hongwei Wu, Resources; David P Giedroc, Conceptualization, Funding acquisition, Writing—original draft

### Author ORCIDs

Daiana A Capdevila (iD) http://orcid.org/0000-0002-0500-1016
David P Giedroc (iD) http://orcid.org/0000-0002-2342-1620

### Decision letter and Author response

Decision letter https://doi.org/10.7554/eLife.37268.050

Author response https://doi.org/10.7554/eLife.37268.051

## Additional files

### Supplementary files

- Supplementary file 1. Supplementary table for wild-type AdcR and selected AdcR mutants zinc binding affinities and melting temperatures from differential scanning fluorimetry.
DOI: https://doi.org/10.7554/eLife.37268.034

- Source code 1. Python script for determining $S^2_{axis}$ values. Representative python script used to analyze sets of peaklists from Sparky (*Lee et al., 2015*) to compute ratios of peak heights and to fit the peak height ratios to η values in order to determine $S^2_{axis}$ values for sidechain methyls of AdcR. Spectral noise must be input for propagation to errors in the peak height ratios, from which Monte Carlo simulation is used to obtain an error estimate for $S^2_{axis}$.
DOI: https://doi.org/10.7554/eLife.37268.035

- Source code 2. Python script for determining relaxation parameters for two state model. Representative python script used to read in a table of peak heights as output from the sparky 'rh' or 'Relaxation Peak Heights' command, where the number of cycles has been entered as the time parameter. This script calculates $R2_{eff}$ for each peak at each number of cycles, then uses these values to fit R2 and $R_{ex}$, given a fixed $\tau_e$, specified in s. Output consists of Excel spreadsheets giving fitted parameters and errors in fitting, as well as PDF charts of each fit and $R_{ex}$ values for all probes.
DOI: https://doi.org/10.7554/eLife.37268.036

- Source code 3. Python script for finding $\tau$ for obtaining relaxation parameters. Representative python script used to read in a table of peak heights as output from the sparky 'rh' or 'Relaxation Peak Heights' command, where the number of cycles has been entered as the time parameter. This script calculates $R2_{eff}$ for each peak at each number of cycles, then uses these values to fit $\tau_e$, R2 and $R_{ex}$ for each probe. Output consists of Excel spreadsheets giving fitted parameters and errors in fitting, as well as PDF charts of each fit and $R_{ex}$ values for all probes.
DOI: https://doi.org/10.7554/eLife.37268.037

- Source code 4. Dynafit script DNA binding isotherms. Representative Dynafit script file used for nonlinear least squares fitting of the fluorescence anisotropy-derived DNA binding isotherms to a dimer linkage model, in which AdcR monomers-dimer equilibrium is displaced to dimers (**P2**) (defined by $K_{dimer} \gg$ protein concentration), which are competent to bind a single AdcO with an affinity Ka. Units of [concentrations] and [constants] in this script are nM and $nM^{-1}$, respectively.
DOI: https://doi.org/10.7554/eLife.37268.038

- Source code 5. Dynafit script $Zn^{II}$-binding isotherms. Representative Dynafit script file used for nonlinear least squares fitting of the $Zn^{II}$-binding isotherms obtained from a titration of apo (metal-free) wild-type AdcR or a mutant AdcR and mag-fura-2 (mf2) with $ZnSO_4$. Units of [concentrations] and [constants] in this script are μM and μM-1, respectively.
DOI: https://doi.org/10.7554/eLife.37268.039

- Transparent reporting form
DOI: https://doi.org/10.7554/eLife.37268.040

### Data availability

NMR datasets were deposited in Biological Magnetic Resonance Bank. Representative data generated or analyzed during this study are included in the supporting files.

The following datasets were generated:

| Author(s) | Year | Dataset title | Dataset URL | Database and Identifier |
|---|---|---|---|---|
| Capdevila D, Edmonds K, Wu H, Giedroc D | 2018 | Backbone and side-chain methyl relaxation rates, methyl order parameters, and stereospecific resonance assignments for Zn(II) AdcR | http://www.bmrb.wisc.edu/data_library/summary/index.php?bmrbId=27447 | Biological Magnetic Resonance Data Bank, 27447 |
| Capdevila D, Ed- | 2018 | Backbone and side-chain methyl | http://www.bmrb.wisc. | Biological Magnetic |

| | | | | |
|---|---|---|---|---|
| monds K, Wu H, Giedroc D | | relaxation rates, methyl order parameters, and stereospecific resonance assignments, for apo AdcR | edu/data_library/sum-mary/index.php?bmrbId=27448 | Resonance Data Bank, 27448 |
| Daiana A Capdevila, Katherine A Ed-monds, Hongwei Wu, David P Gie-droc | 2018 | Backbone assignments and relaxation rates for Zn(II) L57M AdcR | http://www.bmrb.wisc.edu/data_library/sum-mary/index.php?bmrbId=27655 | Biological Magnetic Resonance Data Bank, 27655 |
| Daiana A Capdevila, Katherine A Ed-monds, Hongwei Wu, David P Gie-droc | 2018 | Backbone assignments and relaxation rates for apo L57M AdcR | http://www.bmrb.wisc.edu/data_library/sum-mary/index.php?bmrbId=27656 | Biological Magnetic Resonance Data Bank, 27656 |

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
