## [Decision Letter]

Thank you for submitting your article "Tuning site-specific dynamics to drive allosteric activation in a pneumococcal zinc uptake regulator" for consideration by *eLife*. Your article has been reviewed by three peer reviewers, including Lewis E Kay as the Reviewing Editor and Reviewer #1, and the evaluation has been overseen by John Kuriyan as the Senior Editor. The reviewers have discussed the reviews with one another and the Reviewing Editor has drafted this decision to help you prepare a revised submission.

All three reviewers find the study to be of interest and the role of dynamics and allostery in controlling function in this important system to be important. Yet all three reviewers raise serious concerns, as detailed below, that must be properly addressed in a revised manuscript.

The reviewers raise a number of important points, of which two In particular will require additional experimental work. We view these new data as essential for addressing our collective concerns with the validation of the structure/dynamics/function linkage discussed here. Please let us know if you anticipate problems in responding appropriately to these comments. As you will see, there are several other major points raised by the reviewers, but these are in the nature of requests for clarification or amendment of the manuscript.

Review:

This manuscript by Capdevila et al. details a variety of biophysical studies to examine the allosteric linkage between Zn(II) and DNA binding in the bacterial AdcR transcriptional regulator. This dimeric protein is known to undergo a substantial increase in DNA binding affinity in response to metal binding, but the process by which it does so has remained poorly characterized by crystal structures which have shown only minor conformational changes. In contrast, solution studies, ranging from SAXS to NMR chemical shift perturbation, showed marked metal-induced changes. Backbone and methyl relaxation measurements demonstrated that these structural changes are accompanied by substantial changes in dynamics at several timescales. Of particular note, the authors examined the importance of several methyl-containing residues in the allosteric transmission process, showing those with the greatest changes in S2 (methyl) values upon activation were important in the signal transmission process as assessed by DNA binding measurements. Yet it is felt that some of the claims are based on over-interpretation of the data and that the authors could do a significantly better job in explaining results.

Major points (requiring additional experimental work):

1) To show that the allosteric path is comprised of residues with dynamics the authors make a number of mutations of the dynamic residues (based on sidechain measurements) and show that these affect binding. Yet, while we agree with the importance of dynamics to allostery, we wonder whether the authors are really showing it. What if they take residues in the same general area with DS2=0, would they see a binding affect? They look at 1 control residue far away, but it may be that what is important is changes to residues in a given area, rather than the actual DS2 for that residue. In any event, we do think that the authors should prove it by considering residues with DS2~0 but in the area of interest. We suggest looking at 2-3 residues with DS2~0 and showing that mutation of these has little effect.

2) Do the introduced mutations change the dynamic response to Zn binding that then influences binding to DNA? One might assume so. That is for a sidechain mutant that has a big effect, does one also see a large change in the response to Zn binding at the level of changes in motion of the backbone (say less uncoupling of the DNA and dimerization domains) that would then decrease binding efficacy. We suggest looking at the sidechain mutant with the biggest effect and investigating what happens at the level of the backbone motions that so clearly change in a defined way up binding of Zn.

Other critical points to respond to:

1) The relaxation data are analyzed in a minimalist way and often it is not easy to see what the authors are referring to in their small figures. We suggest major changes to these sections. For example, the authors state that in the Zn2 form the protein tumbles as a complete unit. This is fine, based on the figure. However, they state that the derived correlation time is consistent with expectations from calculations. Please use hydro-NMR or some such program to see what is expected from calculations. It is not clear how the calculations were performed. The authors then state that the DNA binding and dimerization domains have significantly smaller tauc values – 10.5 ns. We find it surprising that both are the same, especially considering the data of Figure 4A were there are very distinct differences in R2/R1 values in certain regions (not all white/not all red). We might have thought that the DNA binding domain would have a smaller tauc than the dimerization domains, given the size difference, if the two were truly uncoupled. Also how clear is it that the apo form is a dimer? The conclusions about how Zn binding decrease fast time-scale motion (NOEs) is difficult to see from the plot. We strongly suggest adding for each a panel that shows R2/R1 or NOE or Rex vs residue with secondary structural elements and loops delineated so that one can see better what appears clear to the authors. We think with the NOE data the authors are trying to say something about taue values, but this is unclear from their description. A more precise and careful discussion is called for here. The authors speak about an increase in sub ns motions in helix 4 upon Zn binding based on NOE values. Yet from the NOE it looks like there is little change to helix 4. The authors then state that there is a concomitant increase in Rex values. How were these obtained (hopefully by CPMG) and again it is very hard to see this from the figures. Panels must be included here as well. Can the CPMG data be fit to a single process?

2) In the last paragraph of the subsection “Zn^II^-induced changes in AdcR conformational plasticity along the backbone” the authors state that the increased internal flexibility of the DNA binding domain (please show J(w) values from a reduced spectral density mapping to better establish the point) may be important for higher DNA binding. Yet why should this be the case? One could easily imagine quite the opposite. That the increased dynamics would be quenched upon DNA binding leading to an entropic penalty. Would it not be better to speak about S2 values for the backbone?

3) The authors state that "unlike backbone.. changes in sidechain S2 values.." We don't agree with this statement. Indeed, it seems like the most compelling argument about dynamics comes from the backbone results that show a very significant change in the overall motion of the DNA binding area from the dimerization domain upon metalation that appears crucial for binding (see Figure 7 that further makes this point).

4) The points made in the text for the sidechain data would be similarly improved by having panels showing S2 and DS2 values vs. residue.

5) Following up on points above. Why do all the mutations decrease binding? Should not some increase it? It is always easy to destroy something via mutation, but shouldn't the model predict that some mutations would improve the allostery or binding. Would one expect that DS2>0 or <0 would have different affects to binding; the data does not suggest so, but we wonder why not?

6) In the Discussion the authors state that upon Zn binding there is a striking redistribution in dynamics onto the DNA binding domain. We don't find it striking really. Perhaps a panel (see above) would help as would more discussion.

7) Figure 7 needs far more discussion. Presumably the DS and DH refer to DNA binding. The introduction of dynamical elements inactivates as there would be a steep entropy price to pay upon binding likely. But then if an organic ligand disrupts binding (bottom, yellow star) why would this affect DS so much. After all there is little dynamics now and so there should not be a penalty upon binding. If the binding of organics affects the ability of the DNA domain to reorient to a proper orientation upon DNA binding one might imagine a predominantly enthalpy affect.

8) The authors analyze over 130 crystal structures of MarR proteins in different states (DNA-binding inactive, DNA-binding active and DNA-bound). This analysis is quite interesting, and provides part of the rationale for investigating the dynamics of a representative member of this family. Table S1 provides a small sample of structures and interprotomer distances for the three states. However, the criteria used to assign each structure to a particular state, particularly the active and inactive states, were not made clear in the manuscript (or in Table S1). Furthermore, it would be useful to provide additional information in the text about how many structures were assigned to each state. The complete dataset could be summarized in a table or in an Excel spreadsheet, including the state that each structure represents, and added to the supplemental data. Table S1 could then be added the main text (as Table 1), as this is a small sample of the entire dataset.

9) Figure 1 summarizes the analysis of the ~130 structures. Overall, it is a very useful figure. However, a few changes might improve it. Panel A should include larger type for the secondary structure and it should be slightly offset from the structure so that the labels are easier to read. Similarly, the zinc ions are small and very difficult to see. Coloring the zinc darker would increase contrast. An energy diagram (Panel B) would be useful, but it is confusing because the "active" and "inactive" states are not made clear. This is similar to the comment made above. Apparently, panel B depicts two hypothetical conformational "states", where the active state (with inducer?) is of higher energy than the inactive state (apo?). Is the y-axis correct? Should it be deltaG or G? If deltaG is correct, what are the two states? Also, why is the DNA-bound state at a higher energy (red shaded region?) then the DNA-free state?

10) Apparently, the conclusion from Figure 1 and Table 1 is that the DNA-bound state has a narrow distribution of conformations and both the allosterically inactivated (DNA-binding inactive by ligand or redox) and active states have broad conformational ensembles. What makes this confusing is that the active and inactive states are not defined. Taken at face value, these data still show that structures determined in the absence of DNA (apparently the active and inactive structures) have a broad ensemble of conformations that in some cases overlap with the DNA-bound conformations. Thus, a similar conclusion might be reached without having to define the active and inactive states, which the authors already indicate is a difficult task (Introduction, third paragraph). In the then end, it is not clear why it is necessary to parse the DNA-free states into active and inactive states, especially with the potential uncertainties of doing this. Despite this limitation, the analysis of the structures still suggests that a conformational ensemble model of allostery might be operative.

11) Figure 4 shows the raw backbone relaxation parameters (R2/R1, hNOE and Rex), while Figure 5—figure supplement 3 shows sample dispersion data for a few residues (NH and methyl groups). The first observation is that the Rex values are relatively small and their corresponding errors large (especially for the NH data). From this figure it is not apparent how significant the NH Rex values are in comparison to the reported error in the fit. Moreover, the quality of data fits (e.g. chi2) are not shown nor discussed in the text (for both NH and methyl groups).

[Editors' note: further revisions were requested prior to acceptance, as described below.]

Thank you for resubmitting your work entitled "Tuning site-specific dynamics to drive allosteric activation in a pneumococcal zinc uptake regulator" for further consideration at *eLife*. Your revised article has been favorably evaluated by John Kuriyan (Senior Editor), a Reviewing Editor, and two reviewers.

The manuscript has been improved but there are some remaining issues that need to be addressed before acceptance, as outlined below.

All three agree that the manuscript is substantially better. There are a number of highlights of the work, including:

1) Analyses of >130 MarR family structures that suggests that their DNA-competent and incompetent states exist as an ensemble.

2) SAXS data, NMR chemical shift data, and NOESY data indicating structural differences between apo and zinc-bound AdcR.

3) Dynamic changes along the backbone and side chain between apo and zinc-bound AdcR (although the backbone changes are subtle, the side chain results are more robust).

4) A correlation between dynamics (DS2axis or DRex) and DGc of zinc-dependent DNA binding.

In particular the identification of dynamic hotspots that are perturbed upon metal binding and the impact that this has on DNA binding is important and of great interest.

In principle the paper is acceptable for publication, but we strongly feel that additional textual changes be made to avoid over-interpretation of the data. In a revised version, considered only at the editorial level, the authors are asked to focus only on 1-4 above and the identification of hotspots that their data nicely show, without additional claims.

Major comments:

1) The main weakness or issue with this paper is that there is no direct evidence for the conclusion that perturbing "dynamically active" residues redistributes the dynamics to impact allostery. The authors make this claim in the revised manuscript (subsection “On-pathway and off-pathway allosterically impaired mutants of AdcR”). Although this claim may very well be true, the backbone data for the L57M variant does not support this hypothesis. Indeed, the authors suggest that the redistribution is likely at the level of side chain dynamics. The alternative possibility (not really described in the Discussion) is that subtle changes in protein conformation (enthalpy) may also be important. In my mind, analyzing the side chain dynamics would be essential for the authors to solidify their claim about the link between dynamical active residues and allostery. Otherwise, the authors should tone down the language and focus on the redistribution of dynamics that their analyses of the WT apo and zinc-bound data already provide.

2) In several places the authors speak about the fact that analysis of crystal structures suggests a conformational selection mechanism for binding. The authors should be careful with this, however, because they do not have kinetic data to support this assumption. Unless the authors can show that a particular binding pathway involving selecting the bound conformation from an ensemble of different conformers is the dominant flux pathway they cannot make this conclusion.

---

## [Author Response]

Major points (requiring additional experimental work):1) To show that the allosteric path is comprised of residues with dynamics the authors make a number of mutations of the dynamic residues (based on sidechain measurements) and show that these affect binding. Yet, while we agree with the importance of dynamics to allostery, we wonder whether the authors are really showing it. What if they take residues in the same general area with DS2=0, would they see a binding affect? They look at 1 control residue far away, but it may be that what is important is changes to residues in a given area, rather than the actual DS2 for that residue. In any event, we do think that the authors should prove it by considering residues with DS2~0 but in the area of interest. We suggest looking at 2-3 residues with DS2~0 and showing that mutation of these has little effect.

In the previous version of the manuscript, we presented the results of two mutants of residues with ∆*S*^2^_axis_ ≈0. We have now characterized an additional three (3) methyl group substitution mutants that are dispersed throughout the structure (Figure 6A and Table S1). In this revised version of the manuscript, we also consider changes in the thermal stability of the mutants as a potential factor that can lead to misinterpretation of the results (Supplementary file 1, Figure 6—figure supplement 4). Indeed, substitution of a number of conserved residues leads to a decrease in thermal stability; we have therefore eliminated from discussion all mutants that are significantly destabilized on the basis of this simple scanning fluorimetry assay.

We also evaluated the proximity of the different control mutant substitutions in the crystal structure. The C_δδ_1-I104 methyl group is less than 5 Å away from the methyl groups of two dynamically active residues we tested previously, in the DNA binding domain (V34 and L36). We expressed and purified V63A AdcR as a possible control substitution in the _αα_3 helix for the dynamically active L57 in the L57M AdcR mutant. However, the thermal stability of this mutant, as well as the L61V AdcR (also in the _αα_3 helix) were found to be significantly lower at physiological pH and thus could not be investigated further. On the other hand, we provide new data on the L17A and I27A substitution mutants that serve as a control residues for the two dynamically active mutants in the dimerization domain (less than 10 Å away from the I16A and V142A substitutions). These have characteristics consistent with expectations from a dynamics- driven allosteric model, i.e.,essentially no impact on allostery.

2) Do the introduced mutations change the dynamic response to Zn binding that then influences binding to DNA? One might assume so. That is for a sidechain mutant that has a big effect, does one also see a large change in the response to Zn binding at the level of changes in motion of the backbone (say less uncoupling of the DNA and dimerization domains) that would then decrease binding efficacy. We suggest looking at the sidechain mutant with the biggest effect and investigating what happens at the level of the backbone motions that so clearly change in a defined way up binding of Zn.

We prepared the V34A and L57M AdcRs for detailed NMR backbone dynamics characterization. ^15^N-TROSY spectra showed that V34A AdcR was not sufficiently stable at 35

°C to obtain backbone assignments in both allosteric states, a finding confirmed by the scanning fluorimetry experiments (Supplementary file 1, Figure 6—figure supplement 4). Thus, we continued with the characterization of L57M AdcR (Figure 6—figure supplements 6 to 11). Much of the structural and dynamic data along the backbone is wild-type-like. This is as expected from the rationale behind this substitution: the L57M AdcR substitution is expected to possess nonnative sidechain dynamics (beyond the scope of this manuscript), accompanied by minor or no perturbations in the structure or dynamics of the backbone, which were measured, as suggested by the reviewer. While we do observe small, local differences in the immediate vicinity of M57 that could contribute to the allosteric impact of the L57M substitutions (Figure 6—figure supplements 7 to 11), these changes are quite minor. We recognize however that side chain dynamics measurements cannot, in and of themselves, rule out any contribution to backbone to allosteric function, and we discuss the L57M mutant in this way (subsection “On-pathway and off-pathway allosterically impaired mutants of AdcR”).

Other critical points to respond to:1) The relaxation data are analyzed in a minimalist way and often it is not easy to see what the authors are referring to in their small figures. We suggest major changes to these sections. For example, the authors state that in the Zn2 form the protein tumbles as a complete unit. This is fine, based on the figure. However, they state that the derived correlation time is consistent with expectations from calculations. Please use hydro-NMR or some such program to see what is expected from calculations.

The residue-specific *R*_1_ and *R*_2_ data obtained for apo- and Zn- AdcRs are now presented in the figure supplements, as requested (Figure 4—figure supplement 1) and compared to the hydroNMR predictions based on the crystal structure of Zn_2_-AdcR at the same temperature and viscosity. This analysis allows us to confirm that the Zn-bound form is tumbling as a single hydrodynamic unit, whereas apo-AdcR has a smaller tumbling time, a finding compatible with more independent domain tumbling in the apo-state. These data are presented along with the hNOE (Figure 4—figure supplement 1A, bottom panel).

We also performed a more comprehensive analysis of these data as suggested by the reviewer, using tensor2. The resulting parameters and the errors of the fitting with the different models are now presented in Figure 4—figure supplement 2 for wild-type AdcR and in Figure 6—figure supplement 10 for L57M AdcR.

It is not clear how the calculations were performed. The authors then state that the DNA binding and dimerization domains have significantly smaller tauc values – 10.5 ns. We find it surprising that both are the same, especially considering the data of Figure 4A were there are very distinct differences in R2/R1 values in certain regions (not all white/not all red). We might have thought that the DNA binding domain would have a smaller tauc than the dimerization domains, given the size difference, if the two were truly uncoupled.

The way in which these calculations were performed are now clarified in the Materials and methods section (subsection “NMR spectroscopy”). We apologize for the confusion we originally raised on the ^ττ _Χ_^ values of the DNA binding and dimerization domains in the apo-state. We were *not* suggesting that the tumbling time of these domains is the same nor did we mean to imply that these two domains tumble entirely independently of one another. This has been clarified on the current version on the manuscript (subsection “Zn^II^-induced changes in AdcR conformational plasticity along the backbone”). The prediction from hydroNMR on these domains is well below the *T*_1_/*T*_2_ ratio obtained for Zn-bound and apo- AdcR; however, the observed average *T*_1_/*T*_2_ ratios from the residues in the DNA binding and dimerization domains in apo-AdcR are in fact intermediate between the independent domain values and the homodimer. These observations suggest that in apo-AdcR, these domains are not fully uncoupled; however, they are not tumbling as a single unit as in the Zn-bound state.

Also how clear is it that the apo form is a dimer?

The oligomerization state of the protein has been independently assessed by gel filtration as shown in Figure 6—figure supplement 2 (done at a lower concentration) and is compatible with the radius of gyration measured from the SAXS analysis (Figure 2, Figure 2—figure supplement 1) Also, a lack of chemical shift differences in residues positioned at the dimer interface reveals that Zn binding does not affect significantly the dimerization equilibrium, at least under these conditions.

The conclusions about how Zn binding decrease fast time-scale motion (NOEs) is difficult to see from the plot. We strongly suggest adding for each a panel that shows R2/R1 or NOE or Rex vs residue with secondary structural elements and loops delineated so that one can see better what appears clear to the authors. We think with the NOE data the authors are trying to say something about taue values, but this is unclear from their description. A more precise and careful discussion is called for here. The authors speak about an increase in sub ns motions in helix 4 upon Zn binding based on NOE values. Yet from the NOE it looks like there is little change to helix 4.

Figure panels that provide all *R*_2_/*R*_1_ (*T*_1_/*T*_2_) and hNOE data have been added in the supplement (Figure 4—figure supplement 1). The description has been clarified and more comprehensive discussion of the *R*_1_, *R*_2_ and hNOE experiments is now presented in the subsection “Zn^II^-induced changes in AdcR conformational plasticity along the backbone”. The ^ττ^
_εε_ values are also now presented in Figure 4—figure supplement 2A-B). Changes on the sub- ns motions in the DNA binding domain are now presented more clearly in the circular-shaped insets to the structure figures in Figure 4C, D.

The authors then state that there is a concomitant increase in Rex values. How were these obtained (hopefully by CPMG) and again it is very hard to see this from the figures. Panels must be included here as well. Can the CPMG data be fit to a single process?

Panels with the parameters obtained from the fitting of the CPMG data are now included in Figure 4—figure supplement 3 and Figure 5—figure supplement 3, for the backbone and side chains respectively. There is no improvement in the fitting if independent parameters are used; we therefore conclude that a single process is sufficient to describe the current data. More details about fitting of the CPMG data is now provided in the Materials and methods section (subsection “NMR spectroscopy”, last two paragraphs).

2) In the last paragraph of the subsection “Zn^II^-induced changes in AdcR conformational plasticity along the backbone” the authors state that the increased internal flexibility of the DNA binding domain (please show J(w) values from a reduced spectral density mapping to better establish the point) may be important for higher DNA binding. Yet why should this be the case? One could easily imagine quite the opposite. That the increased dynamics would be quenched upon DNA binding leading to an entropic penalty. Would it not be better to speak about S2 values for the backbone?

Order parameters on the backbone as well as information about the model used to obtain those order parameters are now presented as figure supplements (Figure 4—figure supplement 2A, B). As the reviewers correctly points out, our interpretation of these data is one of many and in fact, could lead to an entropic penalty upon DNA binding. However, these dynamic changes are not unprecedented and as we have shown for other proteins (Capdevila, 2017; Capdevila, 2018), a particular distribution of site-specific motion can be required to access entropy reservoirs that can facilitate DNA binding.

3) The authors state that "unlike backbone.. changes in sidechain S2 values.." We don't agree with this statement. Indeed, it seems like the most compelling argument about dynamics comes from the backbone results that show a very significant change in the overall motion of the DNA binding area from the dimerization domain upon metalation that appears crucial for binding (see Figure 7 that further makes this point).

We thank the reviewers for encouraging us to perform a more comprehensive estimation of the contribution that backbone dynamics makes to function and particularly to conformational entropy in this system. We have now done that, and have better clarified the contribution of backbone (–*T∆S*_conf_*,_bb_*= 3.5 ± 0.5 kcal mol^-1^) and sidechain (–*T∆S_conf,_*_sc_ = 1.1 ±

0.2 kcal mol^-1^; note that this value has changed slightly from the previous report as a result of a more robust tumbling time obtained from tensor2) dynamics to the entropy of Zn binding. It is interesting to point out that the contribution of sidechains dynamics has been shown to play a much larger role on model systems (Caro, 2017); however, for AdcR, it seems reasonable to suggest that loop dynamics make a more significant contribution to the overall entropy. Also, it should be noted that the estimation of the backbone contribution is based mainly on parameters obtained from molecular dynamics simulations (Sharp, 2015) while the sidechain contribution is supported on a much more extensive experimental analysis (Caro, 2017).

In the current version of the manuscript we emphasize the advantage of measuring the sidechain dynamics to test the predictions of a dynamic model of allostery (subsection “On-pathway and off-pathway allosterically impaired mutants of AdcR”, first paragraph). This is because methyl substitutions have a direct impact on sub-ns sidechain dynamics, since they modify the number of χχ angles which directly determines the contribution to site-specific conformational entropy.

4) The points made in the text for the sidechain data would be similarly improved by having panels showing S2 and DS2 values vs. residue.

We have now incorporated ∆*S*^2^_axis_ values vs. residue as a main text figure (Figure 5A) with the raw *S*^2^_axis_ values vs. residue number for each allosteric state incorporated as a figure supplement to that figure (Figure 5—figure supplement 1).

5) Following up on points above. Why do all the mutations decrease binding? Should not some increase it? It is always easy to destroy something via mutation, but shouldn't the model predict that some mutations would improve the allostery or binding. Would one expect that DS2>0 or <0 would have different affects to binding; the data does not suggest so, but we wonder why not?

The model predicts that a particular (re)distribution of internal motions is necessary for the allosteric regulation. In this case, the allosteric connection means allosteric activation of DNA binding. Our results suggest that the single point mutations introduced in this work perturb the native distribution of internal motions and thus impact to different degrees the allosteric connectivity. We have recently shown that is also the case in other systems (Capdevila, 2017; Glauninger, Chem. Sci., 2017; Capdevila, 2018) and it is the expected result for allosteric hotspots.

On the other hand, it is interesting to speculate what sequence perturbations could potentially enhance the allosteric connection between Zn^II^ and DNA binding. As the reviewer points out, this will require a more challenging experimental approach. We expect that a more complete exploration of the sequence space is necessary to find those sequence perturbations; we are, in fact, currently developing a directed evolution approach to explore those possibilities.

6) In the Discussion the authors state that upon Zn binding there is a striking redistribution in dynamics onto the DNA binding domain. We don't find it striking really. Perhaps a panel (see above) would help as would more discussion.

We apologize for this overstatement; we have removed the word “striking” from this sentence. Now, we explicitly point out what we find striking about this redistribution, particularly the number of residues that change motional regime and are not in close proximity to the Zn binding site.

7) Figure 7 needs far more discussion. Presumably the DS and DH refer to DNA binding. The introduction of dynamical elements inactivates as there would be a steep entropy price to pay upon binding likely. But then if an organic ligand disrupts binding (bottom, yellow star) why would this affect DS so much. After all there is little dynamics now and so there should not be a penalty upon binding. If the binding of organics affects the ability of the DNA domain to reorient to a proper orientation upon DNA binding one might imagine a predominantly enthalpy affect.

Figure 7 has been significantly modified to emphasize the role of conformational entropy while providing a clearer explanation of the concept of entropic reservoirs in the context of allosteric inhibition or activation. We have also thoroughly modified (and significantly shortened!) the text that discusses this figure (subsection “Conclusions”). We do not believe that the orientation of the DNA binding domain necessarily plays a role as large as was inferred by static crystal structures. However, that possibility is now taken into account in the revised Figure 7. We thank the reviewer for these suggestions to improve our presentation of these ideas.

8) The authors analyze over 130 crystal structures of MarR proteins in different states (DNA-binding inactive, DNA-binding active and DNA-bound). This analysis is quite interesting, and provides part of the rationale for investigating the dynamics of a representative member of this family. Table S1 provides a small sample of structures and interprotomer distances for the three states. However, the criteria used to assign each structure to a particular state, particularly the active and inactive states, were not made clear in the manuscript (or in Table S1). Furthermore, it would be useful to provide additional information in the text about how many structures were assigned to each state. The complete dataset could be summarized in a table or in an Excel spreadsheet, including the state that each structure represents, and added to the supplemental data. Table S1 could then be added the main text (as Table 1), as this is a small sample of the entire dataset.

The data in Table S1 is now integrated into Table 1. The criteria have been clarified both in the figure caption and in the footnote of Table 1. In addition, we provide an excel file with all the information available from each crystal structure has been added as additional information.

9) Figure 1 summarizes the analysis of the ~130 structures. Overall, it is a very useful figure. However, a few changes might improve it. Panel A should include larger type for the secondary structure and it should be slightly offset from the structure so that the labels are easier to read. Similarly, the zinc ions are small and very difficult to see. Coloring the zinc darker would increase contrast. An energy diagram (Panel B) would be useful, but it is confusing because the "active" and "inactive" states are not made clear. This is similar to the comment made above. Apparently, panel B depicts two hypothetical conformational "states", where the active state (with inducer?) is of higher energy than the inactive state (apo?). Is the y-axis correct? Should it be deltaG or G? If deltaG is correct, what are the two states? Also, why is the DNA-bound state at a higher energy (red shaded region?) then the DNA-free state?

The suggested changes to panel A of this figure have been made. The terms “active” and “inactive” have been removed from the text, and changed to “DNA binding competent” and “DNA binding incompetent”, respectively. Since the purpose is to summarize a schematic model that could possibly explain Figure 1C, we prefer not to associate at this point “DNA binding competent” and “DNA binding incompetent” with apo and ligand bound, respectively, because this would only be true in ligand-mediated allosterically inactivated MarRs. It is ∆*G* because the unfolded state is the reference state as it is usually done in this kind of energy diagram. The DNA-bound state is not represented; what is shaded is the DNA-binding conformation.

10) Apparently, the conclusion from Figure 1 and Table S1 is that the DNA-bound state has a narrow distribution of conformations and both the allosterically inactivated (DNA-binding inactive by ligand or redox) and active states have broad conformational ensembles. What makes this confusing is that the active and inactive states are not defined. Taken at face value, these data still show that structures determined in the absence of DNA (apparently the active and inactive structures) have a broad ensemble of conformations that in some cases overlap with the DNA-bound conformations. Thus, a similar conclusion might be reached without having to define the active and inactive states, which the authors already indicate is a difficult task (Introduction, third paragraph). In the then end, it is not clear why it is necessary to parse the DNA-free states into active and inactive states, especially with the potential uncertainties of doing this. Despite this limitation, the analysis of the structures still suggests that a conformational ensemble model of allostery might be operative.

We apologize for the confusion. The **“**active” and “inactive” states are now redefined here as “DNA binding-competent” and “DNA binding-incompetent” states and these categories have been clarified in Table 1, and are explicitly stated in the excel file with the appropriate references. Briefly we have classified as follows:

- “DNA binding-competent” is any protein allosteric state that has been shown to bind to DNA in-vitrowith an affinity higher than 10^7^ M^-1^ or is capable of repressing the expression of downstream genes;

- “DNA binding-incompetent” is any protein allosteric state that fails to repress these genes and/or exhibits a significantly lower DNA binding affinity from the DNA binding-competent conformation (at least 10-fold) or an affinity lower than 10^6^ M^-1^;

- Putative DNA binding-competent and incompetent states refer to any protein allosteric state for which the DNA binding properties have not been determined, but the conformational state in the crystal structure is known (i.e., reduced, ligand bound). For these systems is it possible to suggest a DNA binding property (competent or incompetent), particularly when one takes into account the degree of sequence similarity to other MarR repressors.

11) Figure 4 shows the raw backbone relaxation parameters (R2/R1, hNOE and Rex), while Figure 5—figure supplement 3 shows sample dispersion data for a few residues (NH and methyl groups). The first observation is that the Rex values are relatively small and their corresponding errors large (especially for the NH data). From this figure it is not apparent how significant the NH Rex values are in comparison to the reported error in the fit. Moreover, the quality of data fits (e.g. chi2) are not shown nor discussed in the text (for both NH and methyl groups).

The quality of the data fits and the values for all the residues are now discussed in the Materials and methods section. The full Carver-Richards equation was not applied to these data, since we observed no significant improvement in these fits; in fact, the field strength- dependence suggests that all the significant values were not in the slow or intermediate regime.

[Editors' note: further revisions were requested prior to acceptance, as described below.]

Major comments:1) The main weakness or issue with this paper is that there is no direct evidence for the conclusion that perturbing "dynamically active" residues redistributes the dynamics to impact allostery. The authors make this claim in the revised manuscript (subsection “On-pathway and off-pathway allosterically impaired mutants of AdcR”). Although this claim may very well be true, the backbone data for the L57M variant does not support this hypothesis. Indeed, the authors suggest that the redistribution is likely at the level of side chain dynamics. The alternative possibility (not really described in the Discussion) is that subtle changes in protein conformation (enthalpy) may also be important. In my mind, analyzing the side chain dynamics would be essential for the authors to solidify their claim about the link between dynamical active residues and allostery. Otherwise, the authors should tone down the language and focus on the redistribution of dynamics that their analyses of the WT apo and zinc-bound data already provide.

As requested, we have toned down the language regarding the conclusion that perturbation of "dynamically active" residues redistributes the dynamics to impact allostery (subsection “On-pathway and off-pathway allosterically impaired mutants of AdcR”). We agree with the reviewer that we have no direct evidence for this conclusion, which is the subject of ongoing experiments beyond the scope of this paper. We instead use the analysis of the mutants as additional data in support of the conclusions derived from the WT dynamical measurements.

Additionally, we included several sentences in the Results and Discussion section that take into account the uncertainty of the impact of minor structural changes in the functional outcome (subsection “On-pathway and off-pathway allosterically impaired mutants of AdcR”, third paragraph).

2) In several places the authors speak about the fact that analysis of crystal structures suggests a conformational selection mechanism for binding. The authors should be careful with this, however, because they do not have kinetic data to support this assumption. Unless the authors can show that a particular binding pathway involving selecting the bound conformation from an ensemble of different conformers is the dominant flux pathway they cannot make this conclusion.

We agree with the reviewer on this point. The use of the term “conformational selection” in the manuscript has been removed. We now refer exclusively to a conformational ensemble model of allostery (subsection “Conclusions”, first paragraph).